# Feasibility Analysis of Optimal THz bands for passive limb sounding of middle and upper atmospheric wind

Wenyu Wang[1], Jian Xu[1], and Zhenzhan Wang[1]

[1]Key Laboratory of Microwave Remote Sensing, National Space Science Center, Chinese Academy of Sciences, Beijing 100190, China

**Correspondence:** Jian Xu (xujian@nssc.ac.cn)

**Abstract.** As of now, direct measurements of middle and upper atmospheric wind are still scarce, and the observation method is limited, especially for the upper stratosphere and lower mesosphere. This paper presents a study of band selection to derive line-of-sight wind from 30 to more than 120 km using a high-spectral-resolution Terahertz (THz) radiometer, which can fill the measurement gap between lidar and interferometer. Simulations from 0.1–5 THz for evaluating the feasibility of the spaceborne THz limb sounder are described in this study. The results show that high-precision wind (better than $5 \, \mathrm{m \, s^{-1}}$) can be obtained from 40 to 70 km by covering a cluster of strong $O_3$ lines. By choosing strong $O_2$ or $H_2O$ lines, the high-quality measurement can be extended to 105 km. The O atom (OI) lines can provide wind signals in the higher atmosphere. In addition, performance of different instrument parameters include spectral resolution, bandwidth and measurement noise were analyzed and four different band combinations are suggested at last.

## 1 Introduction

Wind is an important meteorological parameter for describing atmospheric dynamical processes. Several studies have already demonstrated that dynamic events occurring in the middle atmosphere have an impact on the tropospheric dynamics and consequently on weather change and climate anomalies in the troposphere (Baldwin and Dunkerton, 2001; Baldwin et al., 2003; Hardiman et al., 2011). The coupling between the atmosphere and ionosphere is greatly influenced by the complex characteristics of the upper stratosphere, and the mesosphere and lower thermosphere (MLT). However, measurements of wind speeds are still lacking in these altitude regions, particularly between 30 and 60 km, which is also known as the "radar gap" (Shepherd, 2015; Hysell et al., 2019; Liu et al., 2023).

Mesosphere, Stratosphere, Thermosphere (MST) radar and meteor radar are typically used in ground-based observation to measure wind above 70 km or below 15 km (Blanc et al., 2018). The upper stratosphere and mesosphere can be measured by Rayleigh-Mie Doppler wind lidar, but only under clear sky conditions (Yan et al., 2017). It has already been suggested to measure the Doppler frequency shift of microwave passive emission to retrieve the wind speed between 70 and 85 km altitude using the CO emission line at 230 GHz (Clancy and Muhleman, 1993). In 2007, a sub-millimeter telescope located in Antarctica at an altitude of 2847 m where the troposphere is very dry, was used to measure CO at 461 GHz to retrieve mesospheric mean winds (Burrows et al., 2007). The first ground-based microwave wind radiometer Wind Radiometer (WIRA) and its successor

Wind Radiometer for Campaigns (WIRA-C) have been able to observe wind profiles between 35 and 75 km altitude for long-term stationary observations from 2012 (Ruefenacht et al., 2012; Rüfenacht et al., 2014; Hagen et al., 2018). In addition, ground-based observations in the 230–250 GHz $O_3$ and CO lines region were also studied in a previous research (Newnham et al., 2016).

The spaceborne interferometers such as Wind Imaging Interferometer (WINDII), Fabry–Perot High Resolution Doppler
Imager (HRDI), and TIMED Doppler Imager (TIDI) were designed to measure wind down to 85, 65 and 70 km, respectively (Gault et al., 1996; Burrage et al., 1996; Killeen et al., 2006). The Atmospheric Laser Doppler Instrument (ALADIN) onboard Aeolus satellite can measure wind below 30 km (Witschas et al., 2020). Using the 118 GHz $O_2$ line measured by microwave radiometer with a high-resolution spectrometer, Microwave Limb Sounder (MLS) onboard Aura satellite first achieved the space microwave measurement of atmospheric line-of-sight wind between 70 and 90 km (a precision of $17\,\mathrm{m\,s^{-1}}$) (Wu et al.,
2008). By measuring the $O_3$ line and HCl line at 625 GHz band, the Superconducting Submillimeter-Wave Limb-Emission Sounder (SMILES) observed wind between 30 and 80 km, with the highest precision being $7\text{-}9\,\mathrm{m\,s^{-1}}$ (Baron et al., 2013b). Its successor SMILES-2 has been proposed to the Japan Aerospace Exploration Agency (JAXA) and wind (30–160 km) is one of the main target parameters (expected precision: $2\text{–}5\,\mathrm{m\,s^{-1}}$) (Ochiai et al., 2017; Baron et al., 2020). Another proposed payload is the Stratospheric Inferred Winds (SIW) instrument of the Swedish InnoSat program which has been selected for
launch in recent years (Baron et al., 2018). Its frequency is centered at 655 GHz and will provide horizontal wind vectors within 30–90 km. THz limb sounder (TLS) is a concept instrument for lower thermospheric neutral wind/density/temperature and can provide wind profiles of 100–180 km (Wu et al., 2016; Yee et al., 2021). A study described by Baron et al. (2013a) provides wind retrieval simulations from 100 GHz to 3 THz, with the bandwidth and the spectral resolution is 8 GHz and 1 MHz, respectively. The conclusion is that the line-of-sight wind can be derived from 25 to more than 90 km with a radiometer
of moderate sensitivity. The vertical resolution is better than 5 km and the precision is better than $10\,\mathrm{m\,s^{-1}}$ in the middle and upper mesosphere (the precision is $2\text{–}3\,\mathrm{m\,s^{-1}}$ from 40 to 65 km). However, investigations on the retrieval performance using a combination of different bands and the impact of instrument parameters were insufficient in the previous studies.

In this study, we present an optimal band selection analysis for measuring wind in terahertz (THz) frequency bands (0.1–5 THz). The sensitivity of wind retrieval to atmospheric molecules at different altitudes and the corresponding frequency bands is an-
alyzed by using radiative transfer simulations. The requirements of spectral resolution, spectral bandwidth and radiometric measurement noise are also discussed. Different band combinations for middle and upper atmospheric wind measurement and the corresponding payload parameters are compared at last. The article is organized as follows. The principle and method are briefly introduced in Sect. 2. The band pre-selection is explained in Sect. 3. Section 4 shows the analysis of different instrument parameters. Section 5 disscusses the limitations of the simulations. At last, Sect. 6 concludes the study.

## 55 2  Methodology

A radiometer with high-resolution spectrometers can measure the Doppler shift of spectral lines caused by the velocity of atmospheric gases with respect to the radiometer. Limb sounding is an effective observation geometry for measuring wind from

space since the emission lines have a much greater signal-to-noise ratio (SNR) and sensitivity of wind than nadir observation. Figure 1 shows the wind measurement principle of a limb sounder. Assuming the spectrum observed by a THz radiometer without the line of sight wind is shown in dark blue and the spectrum with the wind's Doppler shift is shown in red. The difference between these two spectra ($\Delta$BT shown in Fig. 1) is shown in light blue. It can be seen that the wind signature is anti-symmetric and has two position of large spectral differences. Although the value of Doppler shift is small (typically less than 300 kHz) and is difficult to be measured directly due to the system noise at such a high spectral resolution, the variation of brightness temperature ($\Delta$BT) induced by wind can be detected easier. As the sign ale characteristics shown in Baron et al. (2013a), the wind signature is anti-symmetric with respect to the line rest frequency and the position of the maximum BT depends on the line shape but not on the Doppler shift. Therefore, the spectral resolution of measurement does not need to reach as high as 100 kHz and the retrieval problem can be solved by the linear least-squares such as the optimal estimation method (OEM), that is, when the modeled spectrum which considers no wind to compare with the measured spectrum, the wind information can be obtained. It is worth mentioning that the anti-symmetric signature of the wind makes it possible to be retrieved simultaneously with other parameters that have symmetric signatures (such as temperature, pressure, VMR).

## 2.1 Retrieval method

The BT received by the radiometer can be calculated using radiative transfer theory. Neglecting scattering and assuming Local Thermodynamic Equilibrium (LTE), the formal solution of the radiative transfer equation is defined as (Urban, 2003):

$$I_v(S_2) = I_v(S_1)e^{-\int_{S_1}^{S_2} \alpha_v(s)ds} + \int_{S_1}^{S_2} \alpha_v(s)B_v(T)e^{-\int_{S_1}^{S_2} \alpha_v(s)ds}ds, \tag{1}$$

where $I_v$ is the radiance at frequency $v$ from position $S_1$ reaching the sensor $S_2$, $\alpha$ is the total absorption coefficient. $B_v(T)$ stands for the atmospheric emission which is given by the Planck function at temperature $T$.

The OEM is employed as the retrieval algorithm (Rodgers, 2000). The cost function is given by:

$$\chi^2 = [y - \boldsymbol{F}(x,b)]^T \boldsymbol{S}_y^{-1}[y - \boldsymbol{F}(x,b)] + [x - x_a]^T \boldsymbol{S}_x^{-1}[x - x_a], \tag{2}$$

where $y$ is the measurement radiance, $F$ is noiseless radiance calculated from the forward model including instrumental response, $x$ is the target atmospheric state vector, $x_a$ is the a priori state vector, $b$ is the parameters in the model that are independent of the state vector, $\boldsymbol{S}_x$ and $\boldsymbol{S}_y$ are the covariance matrices representing the natural variability of the state vector and the measurement error vector, respectively. The retrieved state vector can be obtained by Gauss–Newton iteration:

$$x_{i+1} = x_a + (\boldsymbol{K}_x^T \boldsymbol{S}_y^{-1} \boldsymbol{K}_x + \boldsymbol{S}_x^{-1})^{-1} \boldsymbol{K}_x^T \boldsymbol{S}_y^{-1}[y - \boldsymbol{F}(x,b) + \boldsymbol{K}_x(x_i - x_a)], \tag{3}$$

where $\boldsymbol{K}_x$ represents the Jacobian matrix of the radiative transfer model.

The retrieval errors can be described by two covariance matrices, the smoothing (or "null-space") error covariance matrix $S_n$ which is due to the a priori profile with an assumed error for regularization:

$$S_n = (A - I)S_x(A - I)^T,$$ (4)

$$A = G_y K_x,$$ (5)

$$G_y = (K_x^T S_y^{-1} K_x + S_x^{-1})^{-1} K_x^T S_y^{-1},$$ (6)

where $A$ is the averaging kernel matrix which represent the sensitivity of the retrieved state to the true state vector, $G_y$ is the contribution function matrix which expresses the sensitivity of the retrieved state to the measurement and $I$ is the unit matrix. The measurement error covariance matrix due to the measurement noise:

$$S_m = G_y S_y G_y^T.$$ (7)

The retrieval error of the following simulations is the total of $S_n$ and $S_m$:

$$\epsilon = \sqrt{diag(S_n + S_m)}.$$ (8)

## 3 Simulations of band selection

### 3.1 Pre-selection of target spectral bands

The spectroscopic lines of common atmospheric molecules in the THz band from the HITRAN-2016 database were used for the line-by-line calculation (Gordon et al., 2017). The following molecules are expected to provide useful wind signals based on their spectroscopic line-strengths and typical Earth's VMR: $H_2O$, $O_3$, CO, $O_2$, HF, HCl and O atom (OI). The simulation atmospheric profiles are derived from the FASCOD standard database, i.e. Air Force Geophysics Laboratory (AFGL) model (Anderson et al., 1986). Profiles cover an altitude range of 0–120 km and include five typical scenarios (15°N, 45°N winter/summer, 60°N winter/summer). The diurnally varying species ($O_3$, NO, and $NO_2$, for example) are estimated from day-side average. Figure 2 shows the five temperature profiles from AFGL and the zonal mean wind profiles from the HWM14 model (Drob et al., 2015).

A simulation of the $\Delta$BT caused by Doppler shift from atmospheric wind was carried out using the Atmospheric Radiative Transfer Simulator (ARTS, v2.4.0) (Buehler et al., 2018). At a frequency interval of 5 MHz, the $\Delta$BTs were simulated at tangent heights of 30, 50, 70, 90 and 110 km. Since all the frequencies are updated for each layer of the atmosphere due to the wind and the absorption coefficients also need to be recalculated. To prevent interpolation errors, the radiative transfer calculation is performed in the ARTS "on-the-fly" mode which means that each absorption coefficient is obtained from instant line-by-line calculation. Five typical profiles including 7 selected molecules mentioned above were used in the pre-selection simulation while only the result of the tropical profile is shown in Fig. 3. The others were calculated but not shown here. Different profiles lead to different $\Delta$BTs, however, the central frequencies are similar.

The results in Fig. 3 demonstrated that BT induced by wind can be as large as 10 K and different molecule lines are sensitive to different altitudes. Although the BT variation (i.e. wind signal) becomes larger with increasing frequency, the system noise temperature also increases correspondingly (see Eq. 9). From the perspective of SNR, most of the high frequencies do not have better SNR than the low frequencies. It can be seen that no signature of winds is noticeable at 30 km because of the pressure broadening of the spectral lines. The number of $\Delta$BT lines at 50 km is the largest and most dense and comes mainly from the numerous $O_3$ spectral lines. The $\Delta$BT at 70 km is significant at the $O_2$ and $H_2O$ spectral lines due to their large VMR or strong line intensity at this altitude region. Furthermore, wind can be obtained up to 90 km by the strong spectral lines of $O_2$ and $H_2O$. Finally, $\Delta$BT at 110 km exists only at 2.06 and 4.74 THz which are the two spectral lines of OI since only this atom has large VMR at this altitude region in THz band molecules. Therefore, based on the above results, those prominent $\Delta$BT positions compared to their surroundings are first selected, such as 118, 448, 487 and 556 GHz lines and so on. Second, other spectral lines that are commonly used or have been mentioned before, even though the $\Delta$BT is not very large, are also taken into account, such as 183, 625 and 773 GHz lines. Third, it is also important to note that since this selection strategy is based mainly on the intensity of the $\Delta$BTs, the $O_3$ line groups with moderate intensity will be missed. Therefore, we have referred to the conclusions of the previous research (Baron et al., 2013a, 2015) such as 359, 655 and 837 GHz line groups and search for the groups of $O_3$ lines with similar density and intensity. The pre-selected central frequencies are listed in Table I.

## 3.2 Performance evaluation

The OEM retrieval was implemented by Qpack2 which uses ARTS as the forward model (Eriksson et al., 2005). Measurements were assumed to scan tangents height from 20 to 120 km at 600 km orbit and to obtain the line-of-sight spectra every 1 km using an antenna, the integration time was assumed to be 0.5 seconds to obtain sufficient system sensitivity. Single side-band (SSB) measurement was assumed in the whole study. The spectral bandwidth and resolution of the radiometer in this simulation was 4 GHz and 1MHz. The system noise temperature for a double side-band (DSB) radiometer at ambient temperature can be simply calculated as a function of frequency (Hubers, 2008):

$$T_{sys} = 50 \times hF/k, \tag{9}$$

where the $h$ is Planck constant, $k$ is Boltzmann constant and $F$ is frequency. The measurement noise or noise equivalent delta temperature (NEDT) of SSB can be calculated as (Baron et al., 2013a):

$$NEDT(SSB) \approx 2 \times NEDT(DSB) = \frac{2 \times T_{sys}}{\sqrt{\beta d\tau}}, \tag{10}$$

where $\beta$ is the spectral resolution (Hz) and $d\tau$ is the integration time (s). The measurement noise can be decreased to half when the radiometer mixer is cooling down to 100 K which is the assumption in the following simulation (i.e. NEDT of SSB multiplied by 0.5).

Due to the purpose of band selection, the a priori molecule profiles which will affect retrieval precision was not considered in this study. The a priori wind profile and the corresponding covariance matrix (used for retrieval regularization) were assumed to be $0 \, \mathrm{m \, s^{-1}}$ and $100 \, \mathrm{m \, s^{-1}}$, respectively. It should be noted that the Zeeman effect of $O_2$ and OI was not considered here, but

it can't be ignored in actual retrieval. Figures 4-8 show the performance of the different frequency bands in wind measurements of the tropical atmosphere. The tropical profile used in this study is typical and this region is important since the QBO (Quasi-Biennial Oscillation) or SAO (Semi-Annual Oscillation) occurs mainly in the tropical troposphere and mesosphere.

According to the above retrieval results, it can be found that the effective range of the radiometer for measuring wind is mostly above 35 km, which may be due to the broadening of the spectral line in the lower atmosphere so that the Doppler signal is hidden in the broadening. Taking the error of $10\,\mathrm{m\,s^{-1}}$ as the limit, the sensitive altitude range of $H_2O$ spectral lines is 45–100 km (see Fig. 4). The 448 and 556 GHz bands show the best performance at 70–80 km and 90–100 km respectively, and bands above 1 THz show larger errors. The 474 and 620 GHz bands are a little better than 448 GHz at 50–70 km while the 620 GHz band errors are large below 50 km. Figure 5 shows the results of $O_3$ bands which are sensitive to 40–85 km. All bands show very high precision at 50–70 km (better than $H_2O$ bands) and the 655 GHz band shows the best performance for stratospheric wind measurement which was already selected for SIW and SMILES-2. Furthermore, the 840 GHz band also show good results in this region and it is possible to cover the 834 GHz $O_2$ line simultaneously to improve the performance at higher altitude. Figure 6 shows that $O_2$ bands are sensitive to 45–100 km which is similar to $H_2O$ bands. However, $O_2$ bands perform better than $H_2O$ bands at 70–90 km due to the nearly constant VMR at these altitudes. The 118 and 487 GHz band shows best performance at this altitude range. The HCl, HF and CO bands show poorer precision compared to other bands (see Figs. 7-8). The OI spectral lines at 2.06 and 4.74 THz show good sensitivity to wind above 105 km which is due to the rapidly increasing VMR of OI in the thermosphere. OI is the only molecule in the THz band that can be used to measure wind at this altitude range. Due to the limitation of the chosen profiles, the simulation was only calculated up to 120 km, the 2.06 and 4.74 THz OI lines have a strong signal above 120 km, for example, the SMLILES-2 is expected to measure wind up to 160 km using the 2.06 THz band. It should also be pointed out that the OI spectral line at 4.74 THz is stronger than that at 2.06 THz, which is suitable for measuring altitudes above 140 km, while the latter is a better choice for lower altitudes. In addition, the $O_3$ lines near these center frequencies also provide information on the stratospheric wind.

Although there are averaging kernels larger than 0.5 at 30–40 km which means the wind is possible to be detected, the precision is poor due to the small SNR. Thus, reducing system noise can improve retrieval precision in the lower stratosphere. Data averaging is also a useful way to obtain the high precision product at this altitude range. Another way is to increase the number of lines with moderate intensity by increasing the bandwidth or using DSB system (Baron et al., 2015).

For wind retrieval, line intensity and shape can both have an impact on $\Delta\mathrm{BT}$. It means that not the stronger spectral line can lead to the higher precision, some less intense lines can also get the better results. It is clear that a steep decrease from the spectral line center to line wings (i.e. narrow line width) is better for wind measurement. According to the results above, using the retrieval error of $5\,\mathrm{m\,s^{-1}}$ as the limit, the altitude is divided into three parts: $\leq 70$ km, 70–100 km and $\geq 100$ km. Considering all molecules in each altitude range, the band with a small retrieval error is preferred, and the lower frequency band is selected under the same conditions. The potential frequency bands include: 118, 448, 487, 556, 655, 840 (broadband can simultaneously cover 834 GHz $O_2$ line) and 2060 GHz.

## 4 Analysis of instrumental parameters

### 4.1 Spectral resolution

Spectral resolution is an important instrument parameter since the Doppler shift from wind is usually small, higher resolution can provide more complete spectrum information but lead to larger measurement noise. Thus, a trade-off between resolution and noise should be considered. Figures 9-15 show the wind retrieval performance assuming the bandwidth is 1 GHz and the spectral resolution is 0.5, 1, 2 and 4 MHz, respectively. Two cases of the "same noise" (i.e. noise from 0.5 MHz resolution) and "normal noise" are compared. It is clear that the higher resolution leads to better retrieval precision under the same noise conditions.

For the 118 GHz band shown in Fig. 9, 0.5 MHz resolution shows good performance at an altitude range of 95–110 km while other resolutions are not sensitive to this range. This is because the narrow Doppler broadening of molecule lines above the stratosphere needs a finer resolution to detect the shift caused by wind. Although 2 MHz resolution shows the best performance below 80 km, the difference is quite small. The effect of resolution is small at the 448 GHz band, and 2 MHz resolution is the best (see Fig. 10). For the 487 GHz band, resolutions of 0.5, 1, and 2 MHz have similar performance while 4 MHz resolution has a large errors increase at 60–80 km which means that this resolution will miss some important spectral line information. Results in Fig. 11 show that 2 MHz resolution is relatively good at the 487 GHz band. The 556 GHz band is sensitive to 70–100 km when the resolution is smaller than 1 MHz, and the 0.5 MHz resolution performs best (see Fig. 12). Results of 655 and 840 GHz bands in Figs. 13-14 show that 1 MHz resolution is best for $O_3$ lines and finer resolution does not bring great improvement, but rather increase the errors in the upper atmosphere. Figure 15 shows that different resolutions have similar errors at the 2060 GHz band above 105 km, and the 2 MHz resolution is finally selected.

### 4.2 Spectral bandwidth

In the upper atmosphere, the Doppler broadening is narrow which indicates that fine spectral resolution is needed as discussed above, while in the lower atmosphere, the large pressure broadening may require a large spectral bandwidth. Figures 16-19 show the wind retrieval performance assuming the resolution is 4 MHz and the spectral bandwidth is 1, 2, 4 and 8 GHz, respectively.

Figure 16 shows the results of two $O_2$ bands, since there are no other strong lines near the 118 GHz band, bandwidth has no impact on the results. It can be found that larger bandwidth improves retrieval errors of 487 GHz band below 80 km. The same conclusion can also be applied to other bands due to the fact that large bandwidth may cover more $O_3$ lines, which provide more wind signals in the lower atmosphere. For the 448 GHz band shown in Fig. 17, increasing bandwidth improves the retrieval errors below 70 km and 4 GHz bandwidth is enough. Fig. 17 shows a bandwidth of 8 GHz can significantly improve the retrieval errors below 70 km at the 556 GHz band. Nevertheless, $O_3$ bands are more appropriate for wind measurement below 70 km. From this perspective, combination with $O_3$ band is a better choice and 2 GHz bandwidth for the $H_2O$ band is sufficient to measure wind at 70–80 km. For the $O_3$ bands depicted in Fig. 18, increasing the bandwidth can significantly improve the retrieval results. Although 4 GHz bandwidth is enough for an altitude above 50 km at 655 GHz band, 8 GHz can

give more information below. Just as the case given by Baron et al. (2013a), the best retrieval results (better than 15 m s$^{-1}$) can be obtained at 25 km using 16 GHz bandwidth at 359 GHz band. It should be noted that an 8 GHz bandwidth of the 840 GHz band can cover the O$_2$ line at 834 GHz which improves the errors a lot above 70 km. Moreover, large bandwidth is no need to

be considered at the 2060 GHz band because this band is only used for wind measurements above 100 km.

According to the simulations above, there are four combinations of different bands selected for atmospheric wind measurement from 0–120 km: 1) 837.5 and 2060.07 GHz; 2) 487.25, 655 and 2060.07 GHz; 3) 118.75, 655 and 2060.07 GHz; 4) 118.75, 448, 655 and 2060.07 GHz. The corresponding parameters are listed in Table II. Although the 556 GHz band was not selected here, it is close enough to the 487 GHz band in terms of measurement altitude and performance to be considered as an

alternative of the 487 GHz band.

### 4.3    System noise temperature

The measurement noise induced by receivers is the main error source in OEM retrieval. System noise temperature and NEDT can be calculated according to the (9) and (10). As stated above, NEDT has a direct impact on measurement precision. This section focuses on the comparison of three different noise cases which is achieved by multiplying the NEDT by different

factors shown in Table II: 1) the radiometer cooled to 100 K (factor is 0.5); 2) the radiometer at ambient temperature (factor is 1); 3) the radiometer with larger noise temperature (factor is 2). The four band combinations discussed above were used in the following simulations.

Figure 20 shows the performance evaluations of the four band combinations with three different noises. It is clear that retrieval error increases with the increase of NEDT. Comparing the different NEDT cases, the radiometer cooled to 100 K

(i.e. NEDT with factor 0.5) shows the best performance and the errors can be better than 5 m s$^{-1}$ in most stratospheric and mesospheric regions and the best is about 1 m s$^{-1}$ at 60 and 100 km. The result of the normal case (i.e. NEDT with factor 1) demonstrates that it is feasible to use a THz radiometer for limb sounding the middle and upper atmospheric wind. The errors in most target altitudes can be better than 10 m s$^{-1}$ and as good as about 3 m s$^{-1}$ at 60 and 100 km. The result of the noisy case (i.e. NEDT with factor 2) is poor but also shows acceptable retrieval errors (better than 15 m s$^{-1}$) from 50 to 100 km.

Considering the case where the NEDT factor is 1 (red line in Fig. 20), Combination 1 shows the poorest performance especially when altitude is above 70 km since the weak O$_2$ line at 834 GHz can't provide sufficient sensitivity to wind at this region. However, with a 10 m s$^{-1}$ error limit, this combination can still measure wind from 45 to about 90 km. The errors are better than 5 m s$^{-1}$ at 50–65 km and about 7 m s$^{-1}$ at 70–85 km. The advantage of Combination 1 is the lowest cost of bands. Combination 2 uses the 487.25 GHz O$_2$ line and 655 GHz O$_3$ lines instead of the 840 GHz band to enhance

the line intensity and reduce the measurement noise for better performance. The errors are better than 6 m s$^{-1}$ from 50 to 95 km. Combination 3 replaces the 487.25 GHz O$_2$ line with 118.75 GHz, further improving the measurement performance above 90 km and increasing the effective altitude to 110 km from 90–100 km in the first two combinations. The errors can be better than 6 m s$^{-1}$ from 50 to 105 km. Although the 118 GHz band has better performance than the 487 GHz band, the disadvantage of this combination is that the 118 GHz radiometer needs a larger antenna and has a lower vertical resolution.

Combination 4 adds an extra 448 GHz band to improve the performance at 60–80 km. Comparing the results of Combination

3 and Combination 4, the retrieval results of 60–80 km have indeed improved, but the improvements are not significant ($\sim 1$ m s$^{-1}$).

## 5   Discussion

From the simulation results shown above, it can be seen that the THz limb sounding has the ability to measure middle and upper atmospheric wind, and the required instrument parameters are not difficult to be achieved. The results of 118 and 487 GHz in Sect. 4.1 suggest that high spectral resolution ($\leq 1$ MHz) can provide more information in the upper atmosphere but this need decreases with increasing frequency. It could be explained by the fact that the line width increases with frequency due to the Doppler broadening. However, the NEDT increases with the increasing frequency and also with the increasing spectral resolution, which also strongly affects the retrieval precision. This needs to be traded off. Furthermore, although the combination of multiple frequency bands is quite beneficial for retrieval precision shown in Sect. 4.3, adding a new band to obtain a small gain should be considered regarding the instrument design complexity and cost.

However, there still are some limitations in this study. Firstly, due to the pre-selecting of relevant molecules, the lines from minor molecules such as NO that could potentially improve wind retrieval are ignored. Secondly, only SSB receiver is considered in this study. The DSB receiver can cover more molecule spectral lines (although the line intensity may become weaker) and provide better results, for example, the 763 GHz band used in SMILES-2 covers the 752 GHz $H_2O$, 773 GHz $O_2$ and NO lines, but the selection of lower and upper bands of DSB is more complicated and not the target of this study. Another limitation is the diurnal changes of $O_3$ and OI which will strongly impact the measurement performance (especially $O_3$ between 60 and 80 km) are not considered here. Baron et al. (2015) presents the retrieval differences between day and night profile and the retrieval performances are degraded in the daytime because of the $O_3$ diurnal variation in the mesosphere.

In addition, the vertical resolution which is not focused in this study needs to be discussed. The antenna size determines the vertical resolution of the instrument in different frequency bands. The larger antenna aperture and the higher frequency will lead to the higher vertical resolution. For example, the THz atmospheric limb sounder (TALIS) (Wang et al., 2020; Xu et al., 2022) has an antenna diameter of 1.6 m, and it has a vertical resolution of about 1 km at 640 GHz and 5.5 km at 118 GHz. The relationship between the antenna size, scanning parameters and vertical resolution can be found in Eq. 6, Baron et al. (2015). This is also the reason why, as mentioned in Sect. 4.3, 487 GHz may be more suitable for satellite observations, even though 118 GHz is more sensitive to wind and has smaller errors above 90 km. The vertical resolution is also related to retrieval error. Finer vertical sampling allows the wind retrieval to be performed at a higher vertical resolution than that of the instrument, but this can result in a big loss of precision. For retrievals with sufficient wind signals, the best compromise between retrieval vertical resolution and precision can be obtained if the retrieval vertical resolution matches the vertical resolution of the instrument (Baron et al., 2015). The horizontal resolution depends mainly on the scanning strategy of the instrument. Large integration time (improve the NEDT), as well as the finer vertical sampling discussed above, will increase the scanning time, leading to poor horizontal resolution.

For the TALIS, the principal prototype developed by National Space Science Center (NSSC), its frequency bands were not designed for wind measurement, so it missed the 655 GHz band and only the 118 GHz band have good wind sensitivity (but the vertical resolution is poor). Based on the results of this study and the miniaturization need of the instrument, the future design of TALIS will primarily consider the 487/556 GHz and 655 GHz bands (2.06 THz is also being developed). Such bands allow for good molecule and wind measurements and obtain ideal vertical resolution at a small antenna size. Since TALIS is a DSB radiometer, future work will optimize the lower and upper side-bands to improve the performance. After the determination of antenna parameters and scanning mode, the effect of vertical resolution will be also considered.

## 6 Conclusions

In this study, simulations of atmospheric wind measurement using a THz limb sounder were performed using radiative transfer model ARTS. The target molecules and corresponding optimal frequencies in the THz band were also analyzed. By comparing the results, $H_2O$, $O_3$, $O_2$ and OI are selected since their spectral lines have good sensitivity to different altitude wind. Although CO, HF and HCl can also provide wind information, their retrieval precision is relatively poor. By combining the best bands of $H_2O$, $O_3$, $O_2$ and OI, good wind measurement (retrieval errors are better than $10 \text{ m s}^{-1}$ and the best is $1 \text{ m s}^{-1}$) can be performed over 40–120 km. The 118, 448, 487, 655, 837 and 2060 GHz frequency bands have been selected for atmospheric wind observation and four combinations of these bands are compared. The 837 and 2060 GHz combination has the advantage of fewer frequency bands and smaller antenna size required to achieve the same vertical resolution, but the observation performance is poorer above 70 km and worse than other combinations due to the larger NEDT at a high frequency. In comparison, the 655 GHz band has a better performance in the stratosphere without losing much vertical resolution. Benefiting from low system noise, 118 GHz band provides high precision retrieval from the mesosphere to the lower thermosphere, but its vertical resolution is too poor for the same antenna size and 487 GHz is a trade-off for this problem. The 2.06 THz is necessary for all combinations.

Spectral bandwidth, resolution and system NEDT of the receiver are the main instrument parameters for radiometric measurement. Due to the spectral line broadening effect, the wind signal is quite weak compared to the linewidth in the lower atmosphere (below 35 km), increasing bandwidth to cover more lines and decreasing NEDT both help improving the retrieval precision. Stratospheric winds can be retrieved with good precision by using $O_3$ lines within a relatively large bandwidth of 4–8 GHz. For MLT retrieval, a small bandwidth of 1–2 GHz is enough. Although Doppler wind measurement requires finer spectral resolution, only the 118 GHz band can provide a significant performance improvement at a resolution of 0.5 MHz due to the negative relationship between resolution and NEDT. With the current considered instrument noise, 1–2 MHz resolution is sufficient. From the simulation results, it is obvious that the radiometer cooled down to 100 K is very important for the high precision observation of upper atmospheric wind. At last, due to the lack of past and future wind observations in the middle atmosphere, using a THz radiometer to fill the needs is a feasible and promising method.

*Code availability.* The ARTS model and Atmlab packages can be obtained at https://www.radiativetransfer.org/ (last access: 12 March 2022).
The HWM14 model is accessible from public repositories (https://doi.org/10.5281/zenodo.240890; Ronald Ilma, 2017)

*Author contributions.* All authors designed the study. WW set up the retrieval experiments, analyzed the retrieval results, and wrote the article. JX provided guidance and insight regarding experiment configurations and results analysis. ZW revised the article.

*Competing interests.* Some authors are members of the editorial board of AMT. The peer-review process was guided by an independent editor, and the authors have also no other competing interests to declare.

*Acknowledgements.* The authors would like to thank the ARTS development teams for assistance and suggestions for the simulations, and also like to thank the editors and the reviewers for their valuable and helpful suggestions.

*Financial support.* This work has been supported by the National Natural Science Foundation of China (grant nos. 42105130 and 42241162) and the China Postdoctoral Science Foundation (grant no. 2021M693201).

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

**Table 1.** Molecules and emission line frequencies for wind measurement

| Molecule | Central Frequency (GHz) |
|---|---|
| $H_2O$ | 183.31, 448, 474.69, 556.93, 620.7, 752.03 (SMILES-2), 916.2, 987.93, 1113.34, 1136.7, 1541.97, 1794.79, 2040.47, 2074.43, 3953.48 |
| $O_3$ | 359, 469, 481.62, 655 (SIW, SMILES-2), 695, 754.46, 840, 866.1, 895.1, 908.7, 1028, 1212, 1396, 1580, 1760, 1943, 2140, 2496, 2651, 2824 |
| CO | 806.65, 1036.91 |
| $O_2$ | 118.75 (MLS), 487.25, 566.9, 715.39, 731.18, 773.84 (SMILES-2), 1058.72, 1061.12, 1406.37, 1466.81, 1525.131, 1711.977, 1812.405, 2214.55, 2502.32, 2846.62, 3190.41, 3469.55, 3811.67, 3930.48, 4153.16, 4271.98, 4900.32 |
| HF | 1232.48, 3691.33 |
| HCl | 625.92 (SMILES), 1251.45, 1876.23, 3121.99, 3736.61, 3742.22, 4353.66, 4360.18, 4975.50 |
| OI | 2060.07 (TLS, SMILES-2), 4744.78 |

**Table 2.** Simulation parameters for different selected bands

| Central Frequency (GHz) | Bandwidth (GHz) | Resolution (MHz) | NEDT (K) Factor: 1 |
|---|---|---|---|
| 118.75 | 1 | 0.5 | 1.13 |
| 448 | 2 | 2 | 2.15 |
| 487.25 | 1 | 2 | 2.34 |
| 655 | 8 | 1 | 4.45 |
| 837.5 | 8 | 2 | 4.03 |
| 2060.07 | 1 | 2 | 9.89 |

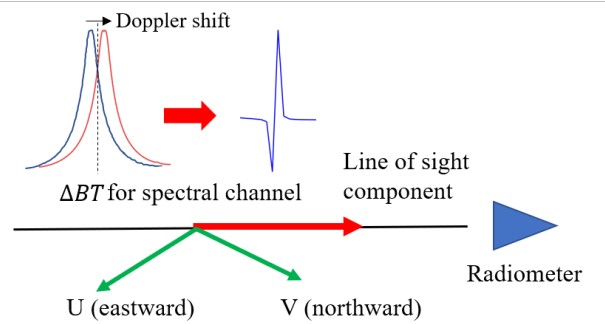

**Figure 1.** Principle of limb sounding of line-of-sight wind.

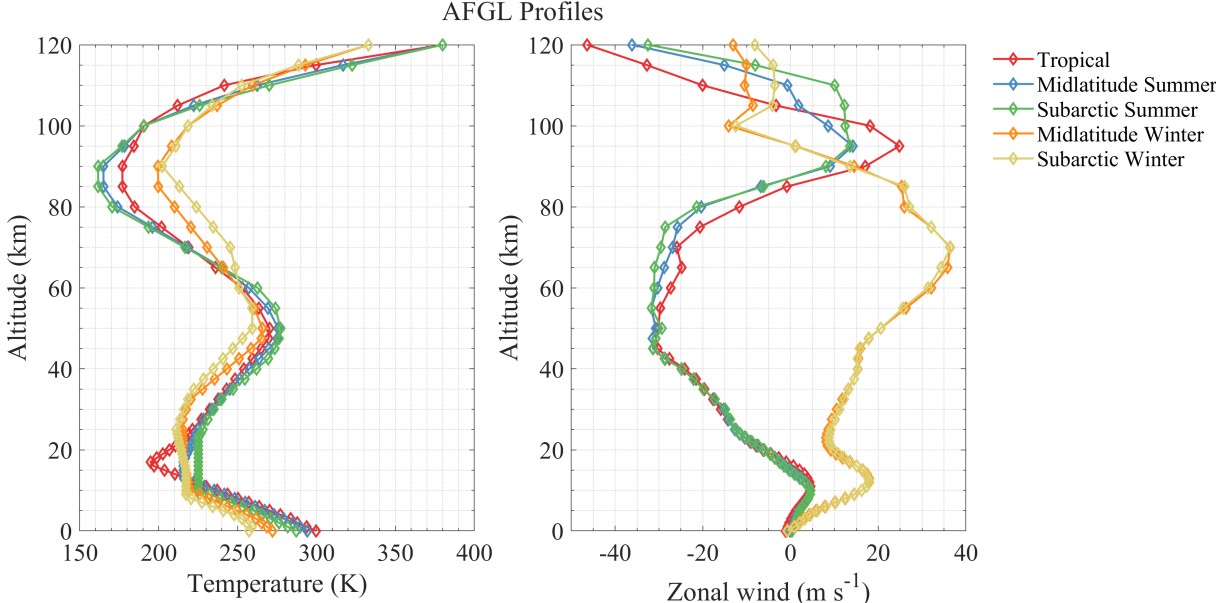

**Figure 2.** Temperature and wind profiles used in simulations.

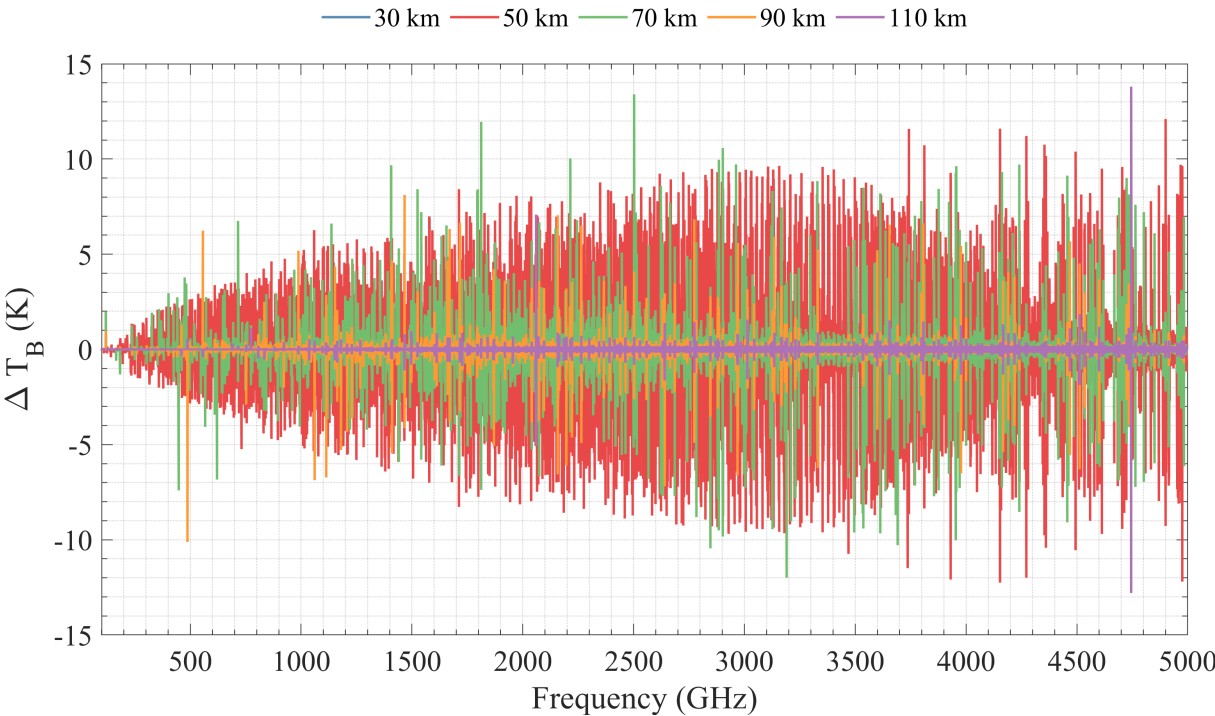

**Figure 3.** ΔBT induced by the wind for tropical profile.

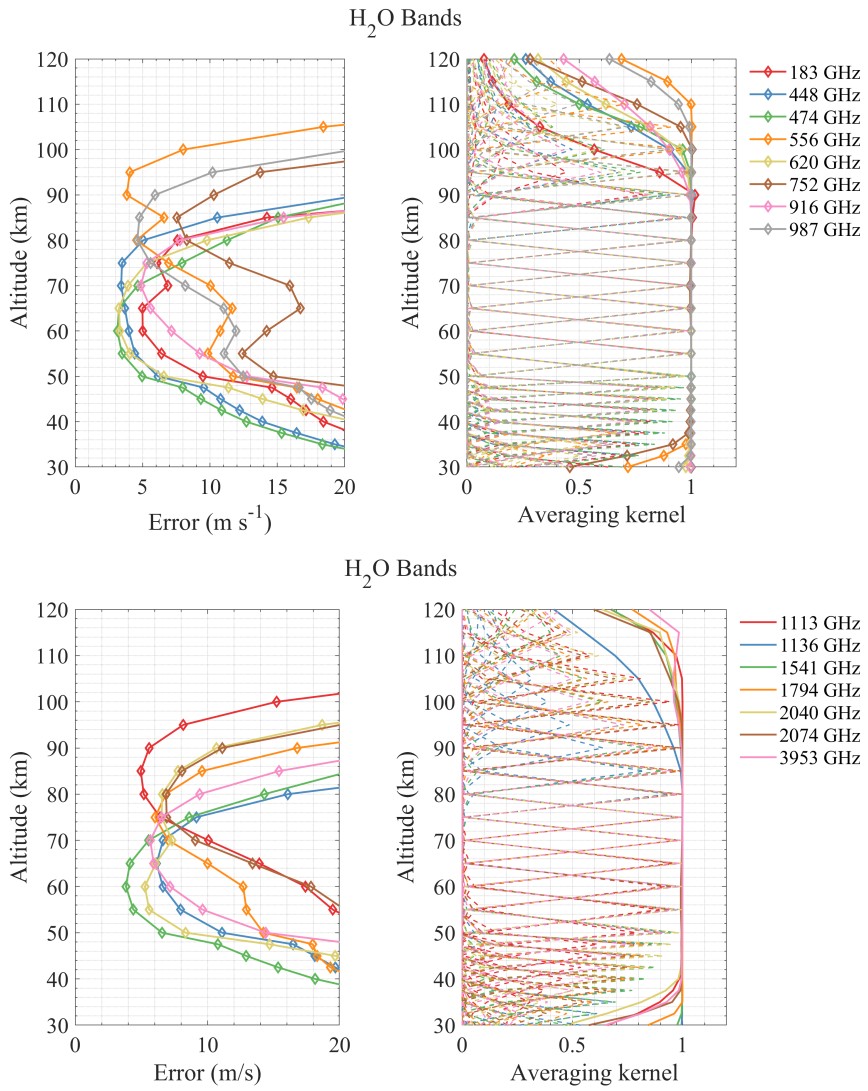

**Figure 4.** Results of wind retrieval using H₂O bands.

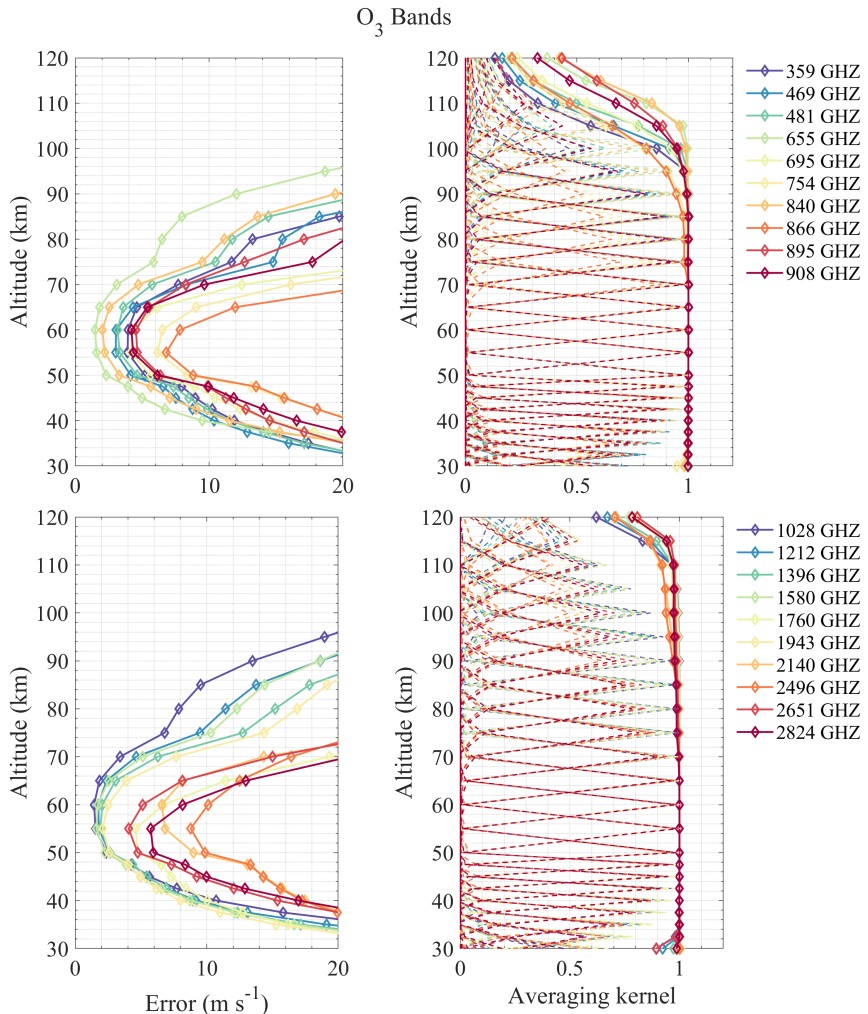

**Figure 5.** Same as Fig. 4 but for O₃ bands.

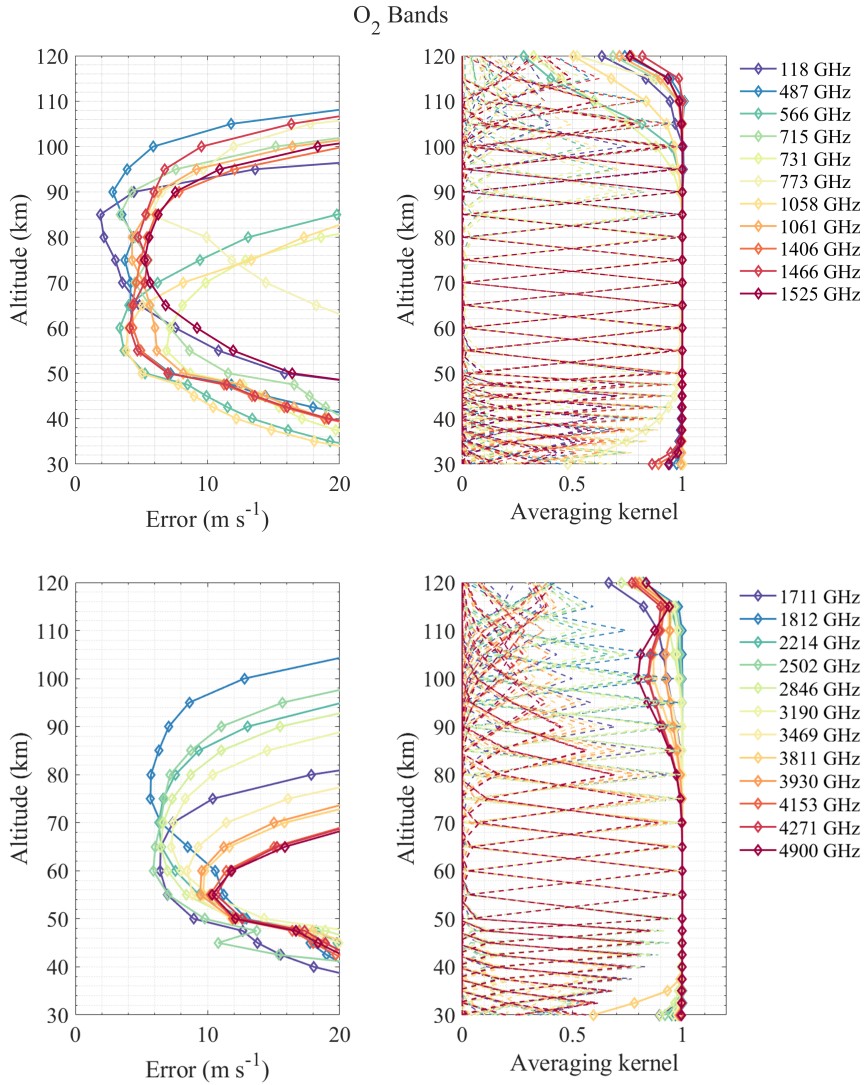

**Figure 6.** Same as Fig. 4 but for O₂ bands.

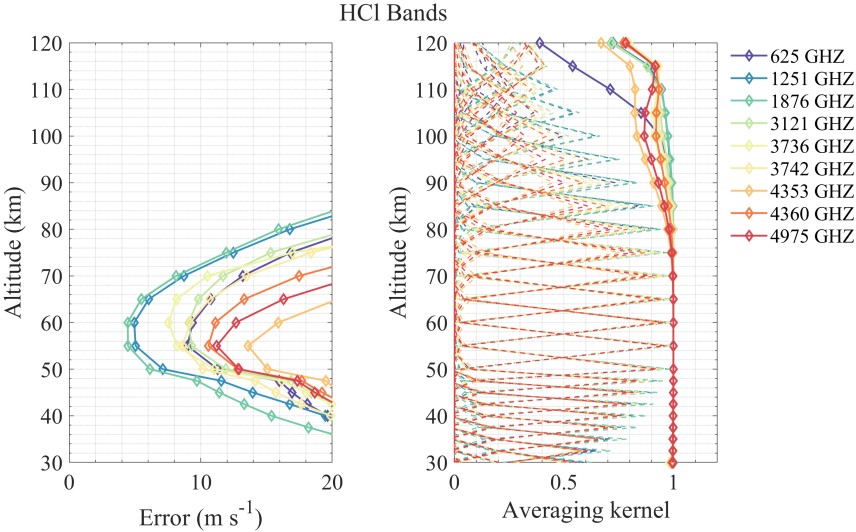

**Figure 7.** Same as Fig. 4 but for HCl bands.

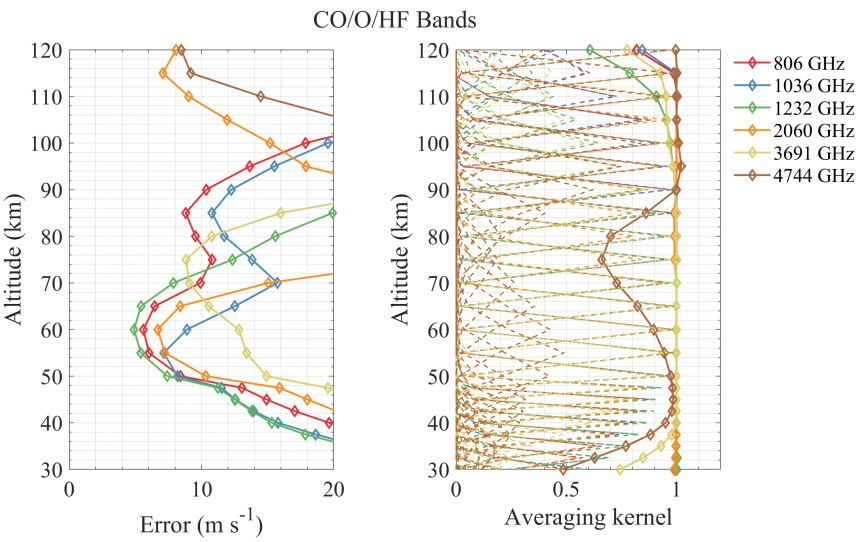

**Figure 8.** Same as Fig. 4 but for CO (806, 1036 GHz), HF (1232, 3691 GHz) and O (2060, 4744 GHz) bands.

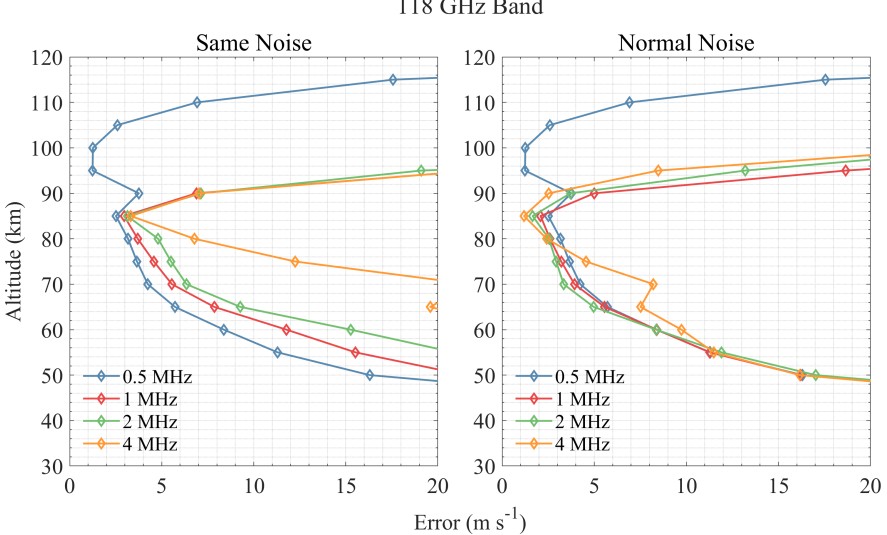

**Figure 9.** Performance comparison of different spectral resolutions of the 118 GHz band.

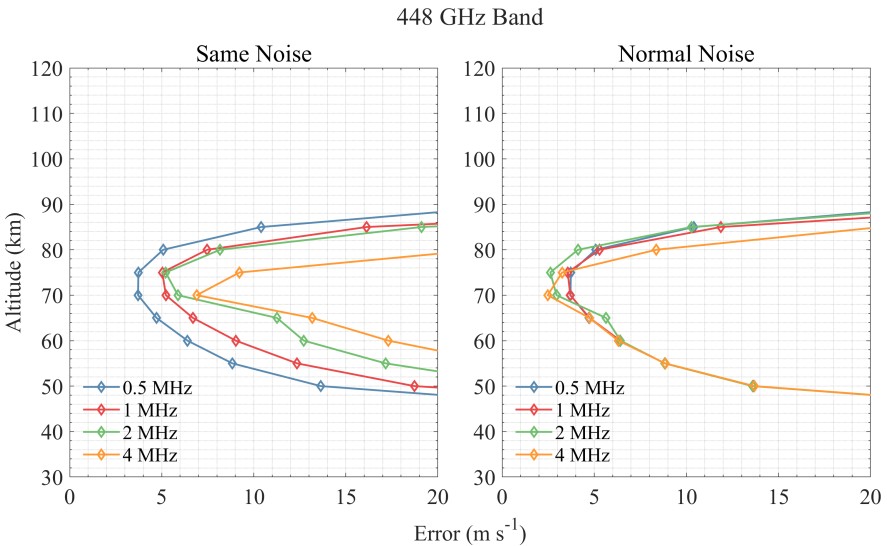

**Figure 10.** Same as Fig. 9 but for the 448 GHz band.

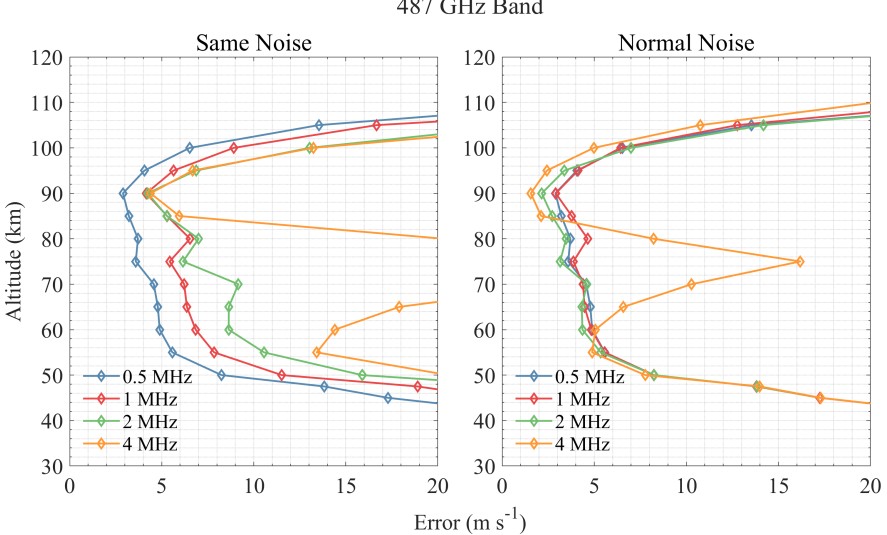

**Figure 11.** Same as Fig. 9 but for the 487 GHz band.

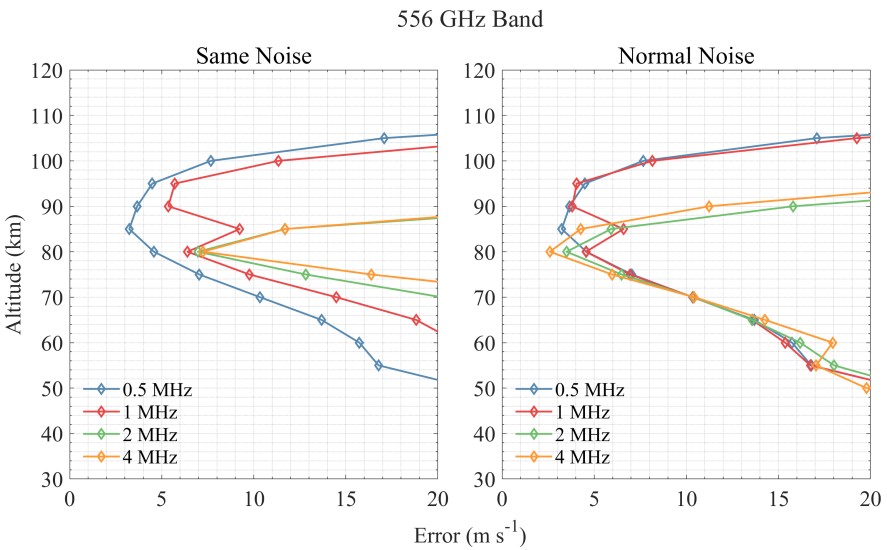

**Figure 12.** Same as Fig. 9 but for the 556 GHz band.

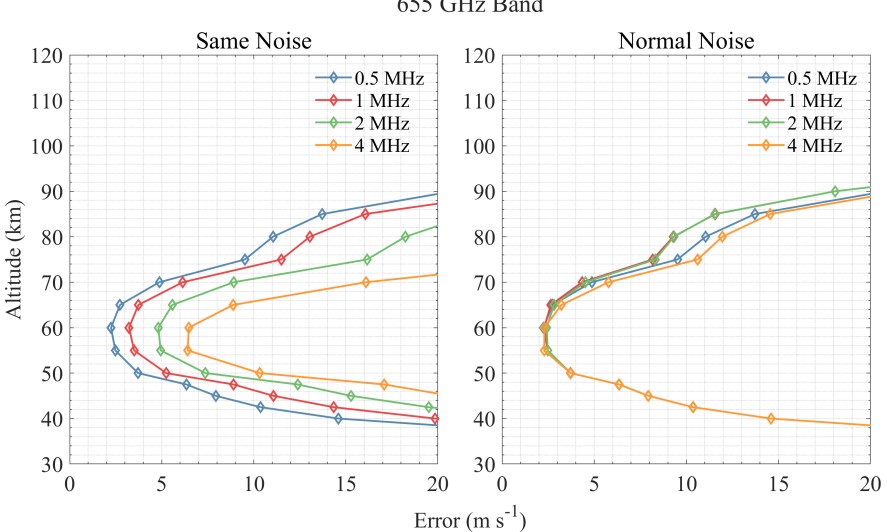

**Figure 13.** Same as Fig. 9 but for the 655 GHz band.

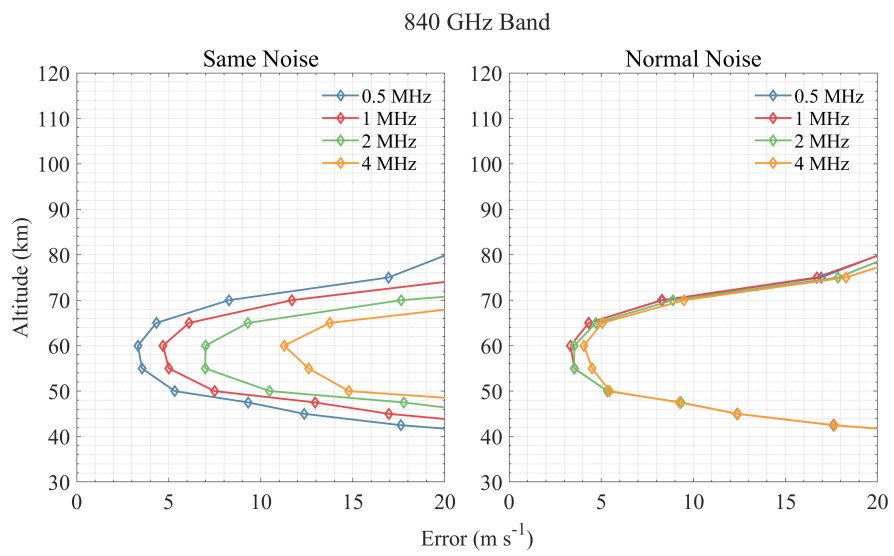

**Figure 14.** Same as Fig. 9 but for the 840 GHz band.

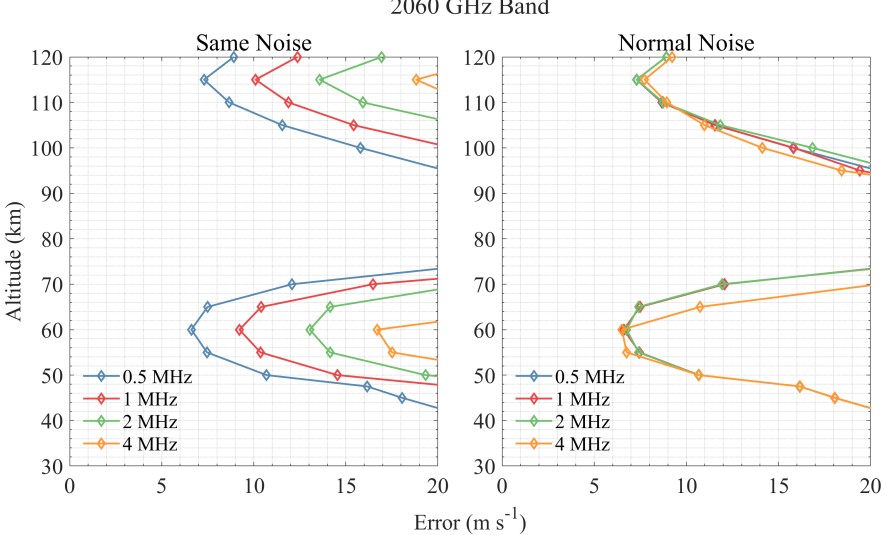

**Figure 15.** Same as Fig. 9 but for the 2060 GHz band.

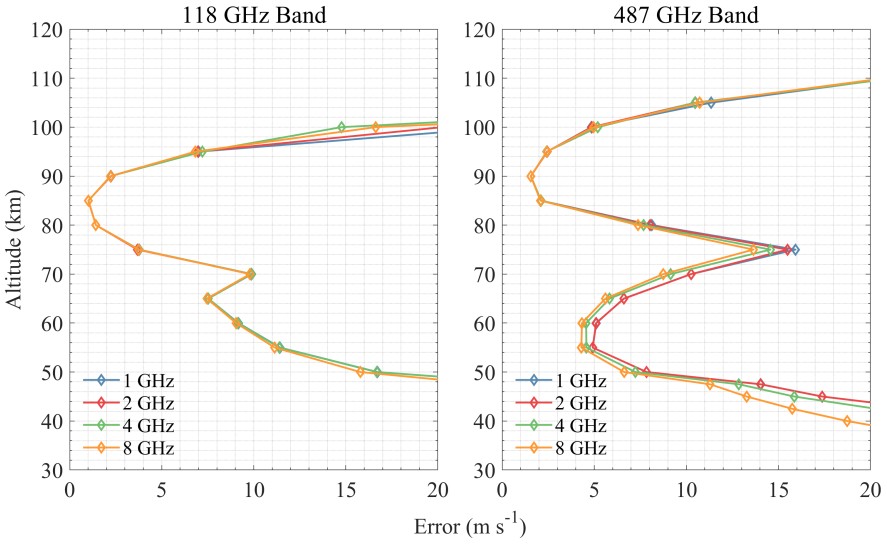

**Figure 16.** Performance comparison of different spectral bandwidths of the 118 and 487 GHz band.

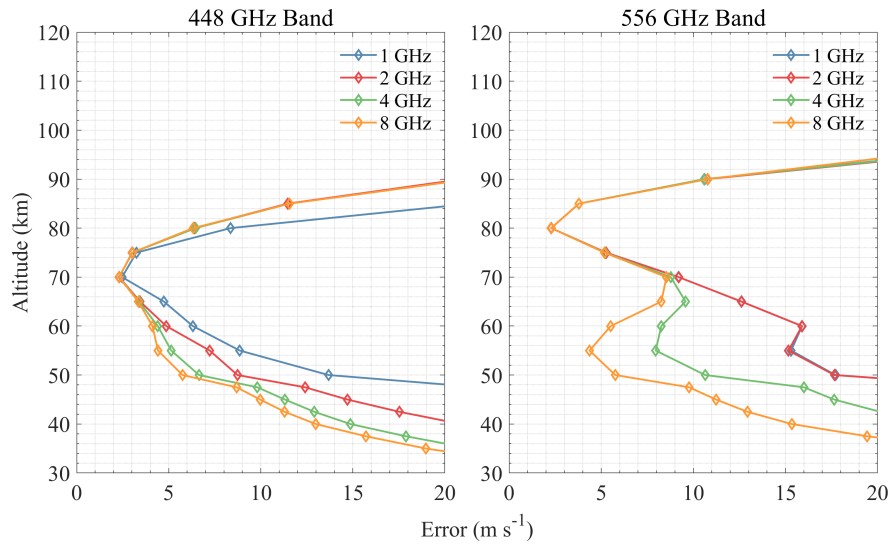

**Figure 17.** Same as Fig. 16 but for the 448 and 556 GHz band.

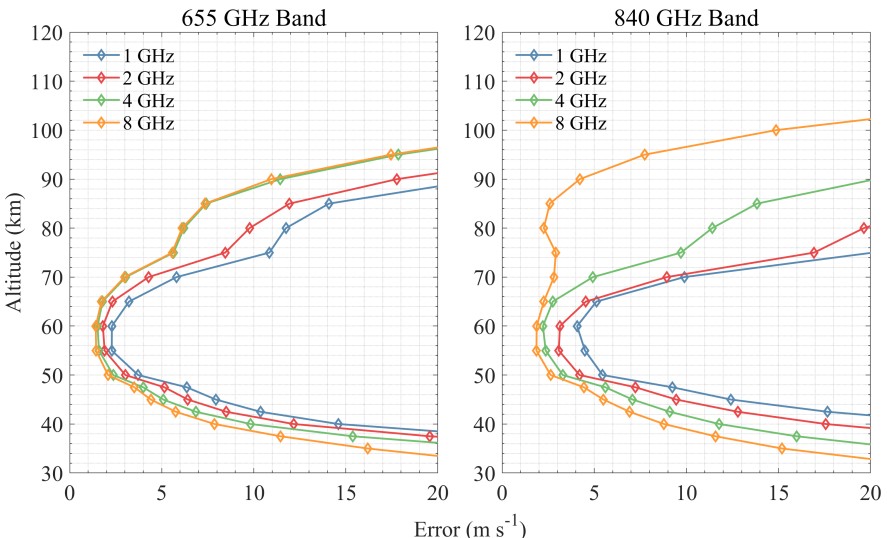

**Figure 18.** Same as Fig. 16 but for the 655 and 840 GHz band.

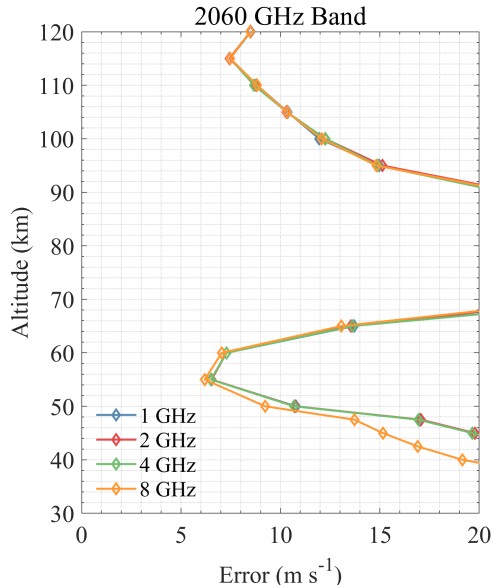

**Figure 19.** Same as Fig. 16 but for the 2060 GHz band.

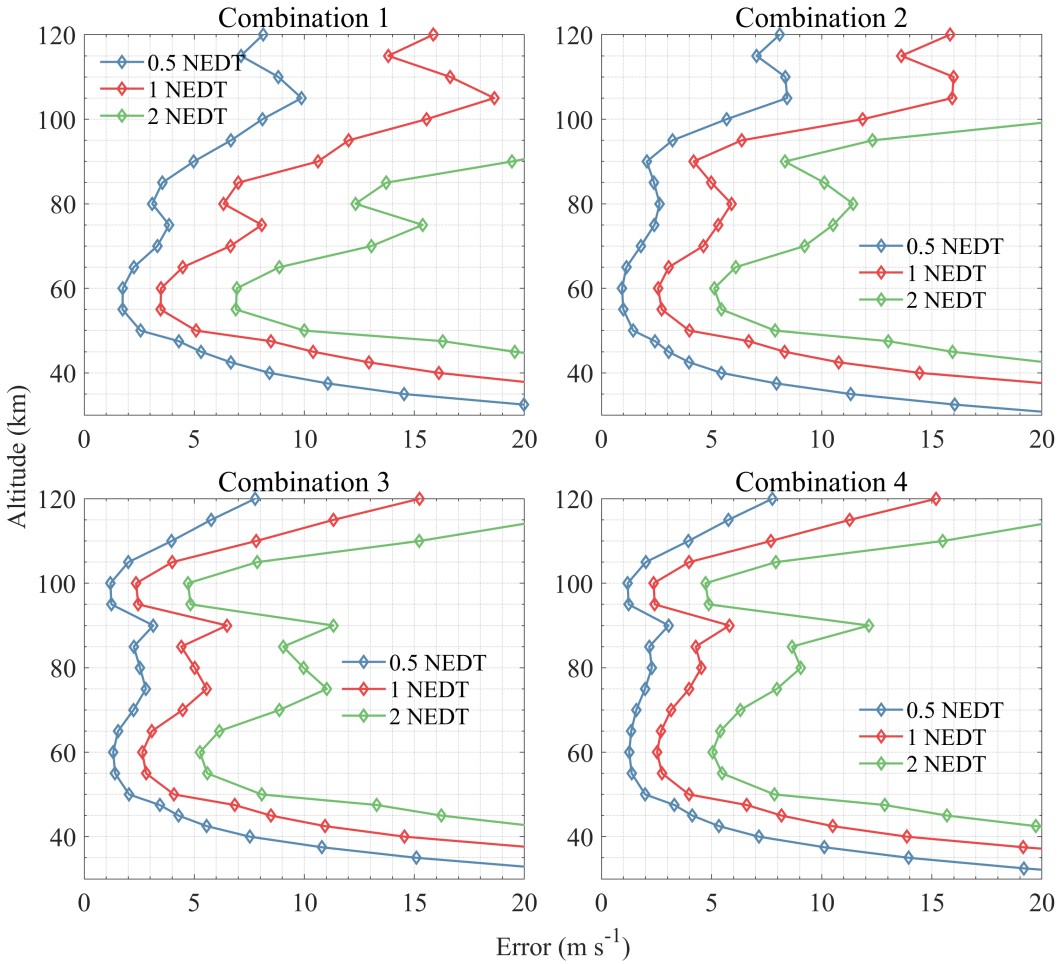

**Figure 20.** Performance comparison of different NEDT of the four band combinations.