# Peer review of "Feasibility Analysis of Optimal THz bands for passive limb sounding of middle and upper atmospheric wind"

_Atmospheric Measurement Techniques, 2023_

## Author Comment (AC1)

We'd like to thank the editor for handling our manuscript, as well as anonymous referee #1 for reading our manuscript carefully and providing numerous professional comments and helpful suggestions. We believe they help us to improve the manuscript significantly and provide many useful ideas for our work.

We have carefully read through all the comments and questions and revised the manuscript accordingly. Please find our point-by-point response to referee #1 below. Here, the reviewer's general and specific questions/comments are formatted in bold font and blue. Our responses are formatted in regular font and black, the manuscript changes are in red.

**The manuscript by Wang et al. presents a sensitivity study to define the best spectral domains to measure middle and upper atmospheric wind with a THz limb sounder. Such analysis complete the few existing works on that matter by considering the most relevant lines over a large frequency domain (100 GHz-5 THz) and the effect of the spectrometer frequency resolution, bandwidth and noise. I believe the study is well suited for AMT, but I have a lot of comments listed below. I believe they will help to clarify issues without major modifications.**

**General comments:**

**The method to select the spectral bands is the key point of this analysis and it is not clear for me. Do the authors utilize Fig. 3 to recognize the individual lines relevant for wind retrieval and adjust the radiometer band for each line to center it? If yes, how do the authors deal with a group of lines with moderate intensity (e.g., 359 GHz)? How did they identify the cluster at 655 GHz?**

Thanks for your comment. It is an important question. As you said, Fig. 3 is the main basis for our choice of frequency bands and of course, this method will miss some groups of lines with moderate intensity which have been proven by Baron et al. (2013, 2015) for good performance of wind measurement. Therefore, we have referred to the conclusions of the previous papers and searched for the groups of $O_3$ lines with similar density and intensity. We have added the description to Line 128: "Therefore, based on the above results, those prominent $\Delta$BT positions compared to their surroundings are first selected, such as 118, 448, 487 and 556

GHz lines and so on. Second, other spectral lines that are commonly used or have been mentioned before, even though the ΔBT is not very large, are also taken into account, such as 183, 625 and 773 GHz lines. Third, it is also important to note that since this selection strategy is based mainly on the intensity of the ΔBTs, the O3 line groups with moderate intensity will be missed. Therefore, we have referred to the conclusions of the previous research (Baron et al., 2013a, 2015) such as 359, 655 and 837 GHz line groups and search for the groups of O3 lines with similar density and intensity".

**The authors chose to conduct their study by pre-selecting relevant molecules. Doing so they ignore lines that could potentially improve wind retrieval, such as strong lines from isotopes of H2O and O2 as well as from minor molecules such as NO. The authors should mention this limitation.**

Thanks for your comment. It is a limitation of this study that should be mentioned in the paper. The isotopes of $H_2O$ and $O_2$ are included in the simulation while the minor molecules such as NO are indeed ignored. We have added the statement to the Scet. 5 (a new discussion section), Line 263: "However, there still are some limitations in this study. Firstly, due to the pre-selecting relevant molecules, the lines from minor molecules such as NO that could potentially improve wind retrieval are ignored. Secondly, only SSB receiver is considered in this study. The DSB receiver can cover more molecule spectral lines (although the line intensity may become weaker) and provide better results, for example, the 763 GHz band used in SMILES-2 covers the 752 GHz $H_2O$, 773 GHz $O_2$ and NO lines, but the selection of lower and upper bands of DSB is more complicated and not the target of this study".

**The authors focus mostly on retrieval errors and do not discuss the vertical and horizontal resolutions. There are trade-off between all these parameters that cannot be overlooked. To account for these issues, the authors should clarify the default observation strategy they assume for these simulations (antenna size, satellite altitude, scan velocity, integration time, …) and could propose possible improvements for future studies. If the default setting is that of TALIS, they could discuss how this study could help to improve its design.**

Thanks for your comment and suggestion. It is an important question in this study. In the Scet. 5, the discussion of vertical and horizontal resolutions and the trade-off which needs to be considered have been added. Since tthe THz limb sounder is proposed to CMA (China meteorological administration) for the next generation of meteorological satellite in the future, the design of TALIS will be modified to accommodate miniaturization needs and the specific parameters such as antenna size have not been decided yet. However, it will be our future work. Thanks again for your suggestion and the references.

The statement of simulations has been rephrased, Line 136: "Measurements were assumed to scan tangents height from 20 to 120 km at 600 km orbit and to obtain the line-of-sight spectra every 1 km using an antenna, the integration time was assumed to be 0.5 seconds to obtain sufficient system sensitivity".

The discussion about antenna, vertical resolution and retrieval has been added to Scet. 5 and given in the specific comment.

Scet. 5, Line 284 describes how this study could help to improve TALIS: "For the TALIS, the principal prototype developed by National Space Science Center (NSSC), its frequency bands were not designed for wind measurement, so it missed the 655 GHz band and only the 118 GHz band have good wind sensitivity (but the vertical resolution is poor). Based on the results of this study and the miniaturization need of the instrument, the future design of TALIS will primarily consider the 487/556 GHz and 655 GHz bands (2.06 THz is also being developed). Such bands allow for good molecule and wind measurements and obtain ideal vertical resolution at a small antenna size. Since TALIS is a DSB radiometer, future work will optimize the lower and upper side-bands to improve the performance. After the determination of antenna parameters and scanning mode, the effect of vertical resolution will be also considered".

**Why the authors have chosen to limit the altitude range to 120 km? The atomic oxygen lines (2 and 4 THz) provide a strong signal above 120 km.**

Thanks for your comment. This is because the AFGL profiles only up to 120 km, we have added a description in Line 169: "Due to the limitation of the chosen profiles, the simulation

was only calculated up to 120 km, the 2.06 and 4.74 THz OI lines have a strong signal above 120 km, for example, the SMLILES-2 is expected to measure wind up to 160 km".

**The diurnal changes of O3 and O are not discussed though they strongly impact the measurement performance (especially O3 between 60-80 km). This should be discussed.**

Thanks for your comment. The discussion has been added to the Scet. 5, Line 268: "Another limitation is the diurnal changes of $O_3$ and OI which will strongly impact the measurement performance (especially $O_3$ between 60 and 80 km) is not considered here. Baron et al. (2015) presents the retrieval differences between day and night profile and the retrieval performances are degraded in the daytime because of the $O_3$ diurnal variation in the mesosphere".

**Specific comments:**

**Line 2: The statement "A new method to derive line-of-sight wind is derived" is not correct since the method described in this study is the same as that used for MLS and SMILES wind measurements.**

Thank you for the correction. It is our mistake and has been revised.

**Line 8: "resolution" should be replaced with "spectral resolution" to not confuse with the spatial one.**

Thanks, it has been revised.

**Line 25: I find the sentence "…have achieved to observe" a bit strange (but I am a non-native English speaker). I would rephrase it as "… have been able to observe wind profiles between 35 and 75 km altitude."**

Thanks, it has been rephrased.

**L27: Same for " in the previously research". It should be "in a previous research."**

Thanks, it has been revised..

**L37: SMILES-2 is designed to measure winds up to 160 km and not 110 km by using the OI line at 2 THz (Ochiai et al. 2017)**

Thank you for the correction. It has been revised.

**Lines 48-50: These sentences are not clear for me. Do they mean that the simulations are performed with TALIS instrumental and observational characteristics? If yes, they should be provided in this manuscript (scan velocity, 1 or 2 LOS?, antenna size, integration time, …)**

We are sorry for the confusion caused here, the sentences here are to present the background of this study is to help improve the frequency band of future versions of TALIS but not use the all current instrument characteristics of TALIS in this simulation. As answered in general comments. This sentence has been moved and discussed in Scet. 5.

**L52: "The sensitivity of atmospheric molecules to wind … "should be "the sensitivity of wind retrieval to atmospheric molecules…"?**

Thanks, it has been rephrased.

**Eq 2: y and F(x) are not defined. It should be stated that F includes instrumental effects, and it is not only Eq 1.**

Thank you for the correction, it has been revised as: "y is the measurement radiance, F is noiseless radiance calculated from the forward model including instrumental response, x is the target atmospheric state vector, $x_a$ is the a priori state vector, b is the parameters in the model that are independent of the state vector" and "where A is the averaging kernel matrix which represent the sensitivity of the retrieved state to the true state and Gy named the contribution function matrix which expresses the sensitivity of the retrieved state to the measurement".

**L89: The sentence could be rephrased as "The retrieval error of the following simulations …". I would use "retrieval error" instead of "precision" since the null-space error component may include systematic errors.**

Thanks, it has been rephrased. The "retrieval error" is used to replace the "retrieval precision" in the paper.

**L64: The statement is not correct "the variation of …induced by wind is amplified by the spectral line broadening". Broader lines make difficult the detection of changes induced by wind. For instance, the lower limit of wind retrieval using THz lines is about 25-30 km because of the large line broadening induced by pressure (as it is explained in L133).**

Thank you for the correction. This is our misrepresentation. What we want to express is that the broadening effect of the spectral lines leads to the specific signatures of the wind which can be detected easier, and that very high spectral resolution is required when the line has very small width. It has been revised as in Line 67: "can be detected easier".

**L69: I think that the following fact is worth mentioning in this paragraph: The anti-symmetric signature of the wind makes it possible to retrieve wind measurements simultaneously with other parameters that have symmetric signatures (temperature, pressure, VMR).**

Thanks for your suggestion. The sentence has been added in Line 71.

**L94-96: The sentence "Since the brightness temperature…." is unclear. A simpler statement such as "The following molecules are expected to provide a useful wind signal based on their spectroscopic line-strengths and typical Earth's VMR: …". Note that because of the saturation of optically thick lines, the statement "the brightness temperature is proportional to … (VMR)" is not always true.**

Thanks for your suggestion. The sentence has been rephrased in Line 105: "The following molecules are expected to provide a useful wind signal based on their spectroscopic line-strengths and typical Earth's VMR: $H_2O$, $O_3$, CO, $O_2$, HF, HCl and O atom (OI)".

**L98: Are the tropical and mid-latitudes profiles an average of day/night conditions?**

According to Anderson et al., 1986, dayside estimates for diurnally varying species ($O_3$, NO, and $NO_2$, for example) were used. This description has been added in Line 109.

**L104: What the "on-the-fly" mode means?**

Thanks for your comment. The "on-the-fly" mode in ARTS means that each absorption coefficient is obtained from line-by-line calculation of spectral data, rather than extracted from a pre-calculated cross-section look-up table. This description has been added in Line 114: "Since all the frequencies are updated for each layer of the atmosphere due to the wind and the absorption coefficients also need to be recalculated. To prevent interpolation errors, the radiative transfer calculation is performed in the ARTS "on-the-fly" mode which means that each absorption coefficient is obtained from instant line-by-line calculation".

**L112: There is no comment for the altitude of 30 km though it is shown in the plot. It could be mentioned that no signature of winds are noticeable at 30 km because of pressure broadening of the lines.**

Thanks for your suggestion. The sentence has been added in Line 123: "It can be seen that no signature of winds are noticeable at 30 km because of pressure broadening of the spectral lines".

**Tab 1: The lines O2@773 GHz and H2O@752 GHz considered for SMILES-2 (Baron et al., 2020) are not selected. It would be interesting to explain why.**

The 752 GHz $H_2O$ and 773 GHz $O_2$ lines have quite strong line intensity, however, with our current setting, their performance is not superior. We think that there are two reasons for this, one is the NEDT, which is about 5 K in our setting, and the second is the SSB receiver used in our simulation. The DSB receiver used by SMILES-2 is better since they combines H2O, O2 and NO lines. This is a limitation of our study and is mentioned in Sect. 5, and it is answered in the general comment. We have added the results of the two bands in Figs. 4 and 6 and labelled the instruments which use these bands in Table 1.

**L115: How the retrieval altitudes are defined?**

The retrieval altitude is the same as the scanning tangent height, from 20-120 km, with 2.5 km intervals for grids below 50 km and 5 km intervals for above.

**L116: What is the size of the antenna. Is-the effect of the scan velocity included in the antenna pattern? What is the integration time (or scan velocity)?**

The antenna is not considered in this study as answered in the general comments. Because it is not the study for a specific payload. The vertical resolution is of course an important parameter that needs to be concerned, and we add the discussion in Sect. 5. The integration time is assumed to be 0.5 seconds to obtain sufficient system sensitivity.

**L126: This sentence is not clear for me. Do you mean that you do not retrieve other parameters than wind velocity?**

Yes, since the main target of this study is to simulate the potential bands, other parameters are assumed to be known and the wind profile is retrieved alone.

**L136: "The 474 … show a little better … " should be rephrased (e.g., "The 474 GHz are a little better …")**

Thanks, it has been rephrased.

**L137 Should be rephrased (e.g., "while the 620 GHz band errors are poorer …")**

Thanks, it has been rephrased.

**L138: Add "s" at "show" in "and the 655 GHz band shows…"**
**It should be indicated that this band was selected for SIW and SMILES-2.**

Thanks, it has been revised in Line 162: "the 655 GHz band shows the best performance for stratospheric wind measurement which was already selected for SIW and SMILES-2". The bands selected by instruments such as SIW, SMILES-2 are also labelled in Table 1.

**L145: It should be stated that the OI@4 THz is much stronger than OI@2 THz: The former is better to measure the altitude above about 140 km while the latter is a better choice for lower altitudes.**

Thanks for your suggestion. It has been rephrased in Line 171: "It should also be pointed out that the OI spectral line at 4.74 THz is stronger than that at 2.06 THz, which is suitable for measuring altitude above 140 km, while the latter is a better choice for lower altitudes".

**L146: "from the stratospheric" -> "on the stratospheric"**

Thanks, it has been rephrased.

**L147: Another way is to increase the number of lines with moderate intensity by increasing the bandwidth or using DSB (Baron et al. [2015]).**

Thanks for your suggestion. The sentence and reference have been added in Line 176.

**L153: I am not convinced by this selection of lines. I don't mean it is not correct, but one needs more explanations: Do you sort automatically the lines by considering all molecules together or molecules by molecules. Which altitude range is considered, ...**

Thanks for your suggestion. Since the same molecule has different sensitive heights in different frequency bands, all molecules are considered together, but the selection is mainly based on the best errors at different altitudes. The sentence has been added in Line 181: "According to the results above, using the retrieval error of 5 m s$^{-1}$ as the limit, the altitude is divided into three parts: $\leq 70$ km, 70–100 km and $\geq 100$ km. Considering all molecules in each altitude range, the band with small retrieval error is preferred, and the lower frequency band is selected under the same conditions".

**L160: The sentence "It is clear …" sounds strange for me. It could be rephrased as: "It is clear that higher resolution leads to better retrieval precision under the same noise conditions."**

Thanks, it has been rephrased.

**L166: "0.5, 1, and 2 MHz resolutions has" -> "resolutions of 0.5, 1, and 2 MHz have"**

Thanks, it has been revised.

**L169: "655 and 840 GHz band" -> "… bands"**

Thanks, it has been revised.

**Sect. 4.1: The results suggest that the need for high resolution (< 1 MHz) for upper-atmosphere wind retrieval decreases with increasing frequency. This could be stated as a general result of this study. It could be explained by the fact that the line width increases with frequency (Doppler broadening).**

Thanks for your suggestion. The sentence has been added to Sect. 5, Line 256: "The results of 118 and 487 GHz in Sect. 4.1 suggest that high spectral resolution ($\leq$ 1 MHz) can provide more information in the upper atmosphere but this need decreases with increasing frequency. It could be explained by the fact that the line width increases with frequency due to the Doppler broadening. However, the NEDT increases with the increasing frequency and also with the increasing spectral resolution, which also strongly affects the retrieval precision. This needs to be traded off. Furthermore, although the combination of multiple frequency bands is quite beneficial for retrieval precision shown in Sect. 4.3, adding a new band to obtain a small gain should be considered regarding the instrument design complexity and cost".

**L176: "Is the targeted spectral line set at the center of the bandwidth?"**

Yes, bandwidth is set at the center of the spectral line.

**L186: "Baron etalBaron et al. (2013)" -> "Baron et al. (2013)"**

Thanks, it has been revised.

**L193: From Baron et al. [2015] it looks like the combination (555,655,2000) is better than (487,655,2000). It would be interesting to explain why you prefer the latter one.**

Thanks for your comment. The article Baron et al. (2015) shows that the 555 GHz has an advantage from 70 to 95 km at some latitudes, and their simulation has a higher spectral resolution of 0.25 MHz. However, this band does not show large advantages in our simulation setting, it may be due to the worse spectral resolution, larger NEDT or the used profiles. Thus, for similar retrieval errors, the $O_2$ line is preferred since its VMR is almost constant, while the

water vapor is affected by changes in atmospheric circulation. The sentence has been added to Line 224: "Although the 556 GHz band was not selected here, it is close enough to the 487 GHz band in terms of measurement altitude and performance to be considered as an alternative of the 487 GHz band".

**Tab 2: I think that it is enough to show a single NEDT (factor 1) since the other values are simply obtained by multiplying or dividing the value by 2 as explained in the text (L198).**

Thanks for your suggestion. It has been revised.

**L218: I think that the issue of the antenna size and the vertical resolution should be discussed a bit more.**

Thanks for your suggestion. The sentence has been added to Sect. 5, Line 271: "In addition, the vertical resolution which is not focused in this study needs to be discussed. The antenna size determines the vertical resolution of the instrument in different frequency bands. The larger antenna aperture and the higher frequency will lead to the higher vertical resolution. For example, the THz atmospheric limb sounder (TALIS) (Wang et al., 2020; Xu et al., 2022) has an antenna diameter of 1.6 m, and it has a vertical resolution of about 1 km at 640 GHz and 5.5 km at 118 GHz. The relationship between the antenna size, scanning parameters and vertical resolution can be found in Eq. 6, Baron et al. (2015). This is also the reason why, as mentioned in Sect. 4.3, 487 GHz may be more suitable for satellite observations, even though 118 GHz is more sensitive to wind and has smaller errors above 90 km. The vertical resolution is also related to retrieval error. Finer vertical sampling allows the wind retrieval to be performed at a higher vertical resolution than that of the instrument, but this can result in a big loss of precision. For retrievals with sufficient wind signals, the best compromise between retrieval vertical resolution and precision can be obtained if the retrieval vertical resolution matches the vertical resolution of the instrument (Baron et al., 2015). The horizontal resolution depends mainly on the scanning strategy of the instrument. Large integration time (improve the NEDT), as well as the finer vertical sampling discussed above, will increase the scanning time, leading to poor horizontal resolution".

**L220: "Whether a new band is needed to improve a little precision still needs to be considered." This sentence is not clear for me. Do you mean: "The benefits of a small gain in retrieval precision by adding a new band should be considered regarding the instrument design complexity and cost"?**

Yes, we are sorry for the unclear statement. The sentence has been rephrased to Sect. 5, Line 260: "Furthermore, although the combination of multiple frequency bands is quite beneficial for retrieval precision shown in Sect. 4.3, adding a new band to obtain a small gain should be considered regarding the instrument design complexity and cost".

**L226: What is the meaning of "total" in "total precision"? I would rather use retrieval error instead of precision (the calculated error includes null-space error which may have systematic errors).**

We are sorry for the confusion statement. The sentence has been rephrased in Line 296: "retrieval errors are better than 10 m s$^{-1}$ and the best is 1 m s$^{-1}$".

**L227: "The 118, 448 … are the final selected…" -> "…have been selected…."**

Thanks, it has been rephrased.

**L228: The statement "for different advantages and disavantages" is too vague.**

Thanks for your comment. The statement has been rephrased in Line 299: "The 837 and 2060 GHz combination has the advantage of fewer frequency bands and smaller antenna size required to achieve the same vertical resolution, but the observation performance is poorer above 70 km and worse than other combinations due to the larger NEDT at a high frequency. In comparison, the 655 GHz band has a better performance in the stratosphere without losing much vertical resolution. Benefiting from low system noise, 118 GHz band provides high precision retrieval from the mesosphere to the lower thermosphere, but its vertical resolution is too poor for the same antenna size and 487 GHz is a trade-off for this problem. The 2.06 THz is necessary for all combinations".

**L229: "of receiver" -> "of the receiver"**

Thanks, it has been rephrased.

**L232: The sentence "The results show …" is not clear. It could be rephrased as: "Stratospheric winds can be retrieved with good precision by using O3 lines within a relatively large bandwidth of 4-8 GHz. For MLT retrieval, a small bandwidth of 1-2 GHz is enough."**

Thanks for your suggestion. It has been rephrased.

**L233 "can derive" -> "can provide" and "very significant" -> "significant" and "at 0.5 MHz" -> "at a 0.5"**

Thanks, it has been deleted.

**L236 The 4K cooled radiometer is not shown in the analysis. I don't think it could be mentioned without additional information.**

Thanks, it has been deleted.

**L238: Why is a THz receiver a superior method? To what is it compared? Measuring middle-atmosphere winds with an IR sounder has been proposed in the past.**

Thanks for your correction. A THz sounder has advantages over optical interferometry and radar in the middle atmosphere. However, we missed the proposed IR sounder, there is very little information about it online. To avoid ambiguity, the "superior" has been rephrased as "promising".

---

## Author Comment (AC2)

We'd like to thank the editor for handling our manuscript, as well as anonymous referee #2 for reading our manuscript carefully and providing numerous professional comments and helpful suggestions. We believe they help us to improve the manuscript significantly and provide many useful ideas for our work.

We have carefully read through all the comments and questions and revised the manuscript accordingly. Please find our point-by-point response to referee #2 below. Here, the reviewer's general and specific questions/comments are formatted in bold font and blue. Our responses are formatted in regular font and black, the manuscript changes are in red.

**General comments:**

**This manuscript presents a sensitivity simulation study to select the optimal frequency bands for middle and upper atmospheric wind measurement using a THz limb sounder. The sensitivity of 0.1-5 THz to wind speed was comprehensively analyzed in a typical profile scenario, and the effects of spectral resolution, bandwidth and system noise were quantitatively analyzed. From my point of view, this sensitivity analysis is meaningful for the specification of future limb sounders and can be suitable for publication in AMT. However, a minor revision is required to clarify the issues that are described below.**

**(1) The first section introduced the past and planned radiometers for measuring wind, but the relevant frequency bands were not described explicitly later in the manuscript. More discussion or analysis should be provided.**

Thanks for your suggestion. We have labelled the instruments which use these bands in Table 1. We also added the statement in the paper when the similar band is selected in our simulation, such as: "the 655 GHz band shows the best performance for stratospheric wind measurement which was already selected for SIW and SMILES-2".

**(2) As stated in referee report #1, the vertical resolution that is a critical parameter for satellite observations, has not been discussed throughout the manuscript, which should be included in the revised manuscript.**

Thanks for your suggestion. The sentence has been added to Sect. 5, Line 271: "In addition,

the vertical resolution which is not focused in this study is needs to be discussed. The antenna size determines the vertical resolution of the instrument in different frequency bands. The larger the antenna aperture and the higher the frequency, the higher the vertical resolution. For example, the THz atmospheric limb sounder (TALIS) (Wang et al., 2020; Xu et al., 2022) has an antenna diameter of 1.6 m, and it has the vertical resolution of about 1 m at 640 GHz and 5.5 m at 118 GHz. The relationship between the antenna size, scanning parameters and vertical resolution can be found in Eq. 6, Baron et al. (2015). This is also the reason why, as mentioned in Sect. 4.3, 487 GHz may be more suitable for satellite observations, despite the fact that 118 GHz is more sensitive to wind and has smaller errors above 90 km. The vertical resolution is also related to retrieval error. Finer vertical sampling allows the wind retrieval to be performed at a higher vertical resolution than that of the instrument, but this can result in a big loss of precision. For retrievals with sufficient wind signals, the best compromise between retrieval vertical resolution and precision can be obtained if the retrieval vertical resolution matches the vertical resolution of the instrument (Baron et al., 2015). The horizontal resolution depends mainly on the scanning strategy of the instrument. Large integration time (improve the NEDT), as well as the fine vertical sampling discussed above, will increase the scanning time, leading to poor horizontal resolution".

**Specific comments:**

**Sect. 2, Line 64: Why "the variation of brightness temperature (BT) induced by wind is amplified by the spectral line broadening and can be detected"? How the doppler shift can be obtained using a spectral resolution that is larger than the doppler frequency shift?**

Thanks for your comment. This is our misrepresentation. What we want to express is that the broadening effect of the spectral lines leads to the specific signatures of the wind which can be detected easier, and that very high spectral resolution is required when the line has very small width. It has been revised as: "can be detected easier". For the doppler shift, we are obtaining it by an indirect method, which is the $\Delta$BT in Fig. 1. The Doppler shift causes the spectral lines to appear to vary antisymmetrically, and this information can be obtained even using a resolution of a few MHz.

**Sect. 3.1, Line 105: What are the differences in simulation results between these typical profiles? Do these differences affect the final conclusion?**

Different season or latitude profiles show differences in retrieval performance but it will not affect the selection of potential bands. Bands in Table 1 were selected from five AFGL profiles and the sensitive bands were similar for different profiles. The tropical profile used in study is typical and this region is important since the QBO (Quasi-Biennial Oscillation) or SAO (Semi-Annual Oscillation) occurs mainly in the tropical troposphere and mesosphere. There are certain effects of different profiles in retrieval errors, such as molecules with diurnal variation. We have added the statement to the Scet. 5, Line 268: "Another limitation is the diurnal changes of $O_3$ and OI which will strongly impact the measurement performance (especially $O_3$ between 60 and 80 km) is not considered here. Baron et al. (2015) presents the retrieval differences between day and night profile and the retrieval performances are degraded in the daytime because of the $O_3$ diurnal variation in the mesosphere".

**Sect. 3.1, Line 103: Where is the BT at tangent heights of 20 km in Fig. 3?**

We are sorry for the mistake here, the tangent heights in Fig. 3 are 30, 50, 70, 90 and 110 km, there is no 20 km here.

**Sect. 3.1, Line 113: How did you select the spectral lines from Fig. 3? It appears that some of the frequencies in Table 1 are not apparent in Fig. 3.**

Thanks for your comment. The $\Delta$BTs in different bands from Fig. 3 is the main reference of selection and the bands with large $\Delta$BTs will be selected. However, this method will miss some groups of lines with moderate intensity which have been proven by Baron et al. (2013, 2015) for good performance of wind measurement. Therefore, we have referred to the conclusions of the previous papers and searched for the groups of $O_3$ lines with similar density and intensity, this is the reason some of the frequencies in Table 1 are not apparent in Fig. 3. We have added the description to Line 131: "It is important to note that since this selection strategy is based mainly on the intensity of the $\Delta$BTs, the $O_3$ line groups with moderate intensity will be missed. Therefore, we have referred to the conclusions of the previous studies (Baron et al., 2013a, 2015) such as 359 GHz, 655 GHz, 837 GHz line groups and search for the groups

of $O_3$ lines with the similar density and intensity".

**Sect. 3.2: For O3 lines at 1028 GHz the retrieval error seems acceptable, why is this frequency not selected?**

It is true that 1028 GHz band also show good retrieval performance. However, it does not show better performance than the 655 GHz, and for the same performance we give preference to the lower bands because the instruments are easier to implement. The sentence to describe the selection strategy has been added in Line 180: "According to the results above, with the retrieval error of 5 m s$^{-1}$ as the limit, the altitude is divided into three parts: $\leq 70$ km, 70–100 km and $\geq 100$ km. Considering all molecules in each altitude range, the band with small retrieval error is preferred, and the low frequency band is selected under the same conditions".

**Sect. 4.1: Why does the 0.5 MHz spectral resolution play such a big role in 118 GHz retrieval, while other bands not?**

This is because the Doppler broadening in the upper atmosphere is small and high resolution (< 1 MHz) is needed for wind retrieval. However, the need decreases with increasing frequency due to line width increases with frequency. The lower frequency and lower system noise at 118 GHz compared to other high frequency bands allows for better performance with higher spectral resolution. The sentence has been added to Sect. 5, Line 256: "The results of 118 and 487 GHz in Sect. 4.1 suggest that high spectral resolution (<= 1 MHz) can provide more information in the upper atmosphere but this need decreases with increasing frequency. It could be explained by the fact that the line width increases with frequency due to the Doppler broadening. However, the NEDT increases with increasing frequency and also with increasing spectral resolution, which also strongly affects the retrieval precision. This needs to be traded off".

**Sect. 5: As we know, interferometers are mainly used for middle and upper atmospheric wind field measurements. , What would be the advantages and disadvantages of using THz limb soundersthen? Please expland the relevant discussion.**

As we discussed in Sect. 1, the THz limb sounder has good performance in measuring

middle atmosphere winds, and less affected by diurnal variations. The THz limb sounder, interferometer, and radar/lidar can form a comprehensive measurement program for full-altitude atmospheric wind measurement.

---

## Author Comment (AC3)

We'd like to thank the editor for handling our manuscript, as well as anonymous referee #3 for reading our manuscript carefully and providing numerous professional comments and helpful suggestions. We believe they help us to improve the manuscript significantly and provide many useful ideas for our work.

We have carefully read through all the comments and questions and revised the manuscript accordingly. Please find our point-by-point response to referee #3 below. Here, the reviewer's general and specific questions/comments are formatted in bold font and blue. Our responses are formatted in regular font and black, the manuscript changes are in red.

**General comments:**

**This manuscript uses the radiative transfer model and the optimal estimation algorithm to simulate the inversion of atmospheric wind speed for terahertz limb sounding, screen out the target molecules in the THz band and the corresponding optimal frequencies, and analyze the influence of spectral bandwidth, resolution and NEDT on the inversion accuracy. But I have the following questions, which need to be revised and answered:**

- **In "As of now, direct measurements of middle and upper atmospheric wind are still scarce" and "However, measurements of wind speeds are still lacking in these altitude regions, particularly between 30 and 60km, which is also known as the "radar gap"", could some more recent references be added here?**

Thanks for your suggestion. Two references have been added which described the "radar gap" and the challenge of measuring winds in the stratosphere and lower mesosphere.

Reference:

Hysell, D. L., Chau, J. L., Coles, W. A., Milla, M. A., Obenberger, K., and Vierinen, J.: The Case for Combining a Large Low-Band Very High Frequency Transmitter With Multiple Receiving Arrays for Geospace Research: A Geospace Radar, Radio Science, 54, 533–551, 2019.

Liu, X., Xu, J., Yue, J., and Andrioli, V. F.: Variations in global zonal wind from 18 to 100\,km due to solar activity and the quasi-biennial oscillation and El Niño–Southern Oscillation during 2002 2019, Atmospheric Chemistry and Physics, 23, 6145–6167, 2023.

- **For "and the observation method is limited", compared with existing equipment, what are the advantages of space THz equipment?**

As we discussed in Sect. 1, the THz limb sounder has good performance in measuring middle atmosphere winds (especially 40-70 km), and less affected by diurnal variations. The current lidar such as Aeolus/ALADIN measures wind below 30 km, interferometer such as ICON/MIGHTI measured wind above 90 km, TIMED/TIDI measured wind above 70 km during the day and 80–105 km at night. The THz sounder can be a supplement to current space measurement.

- **In "THz limb sounder (TLS) is a concept instrument for lower thermospheric neutral wind/density/temperature and can provide wind profiles of 100–180km", can a brief description of the pros and cons of this THz instrument be made?**

It is designed for small satellite platforms with very small mass (<3-5 kg) and power (<30 W). It can measures vector winds from a combined 45° and 135° views with wind speed uncertainty of <10 m/s. However, due to its DSB noise temperature of 7000 K, 10 s integration time is needed to obtain better measurement sensitivity. Thus, its horizontal resolution is 400 km and vertical resolution for two bands is 3.5 km and 6.4 km which are relatively bad but can meet the basic scientific needs.

- **In "However, investigations on the retrieval performance using a combination of different bands and the impact of instrument parameters were insufficient in the previous studies." 。please briefly talk about where the previous research is insufficient and why it is important.**

Thanks for your comment. The previous research were mainly carried out by Baron et al., 2013, 2015. Their work investigated the performance of several focal frequencies and instrument configurations for wind measurements, but they were mainly oriented towards the design of SMILES-2. This study based on previous researches and focuses on more potential frequencies and combinations for providing a generalized reference for future instrument design.

- **For "Figure 1 shows the wind measurement principle of a radiometer", it is necessary to briefly describe the detection principle and advantages, and further, explain the**

**diagram in Fig.1.**

Thanks for your suggestion. The explaination has been added: "Assuming the spectrum observed by a THz radiometer without the line of sight wind is shown in dark blue and the spectrum with the wind's Doppler shift is shown in red. The difference between these two spectra ($\Delta$BT shown in Fig. 1) is shown in light blue. It can be seen that the wind signature is anti-symmetric and has two position of large spectral difference" and "…that is, when the modeled spectrum which considers no wind to compare with the measured spectrum, the wind information can be obtained. It is worth mentioning that the anti-symmetric signature of the wind makes it possible to be retrieved simultaneously with other parameters that have symmetric signatures (such as temperature, pressure, VMR)".

- **For "Therefore, the spectral resolution of measurement does not need to reach as high as 100kHz and the retrieval problem can be solved by the linear least-squares method", maybe there is a mistake here, the optimal estimation algorithm is used, not the least square method.**

Thanks for your comment. However, the optimal estimation algorithm is also one of the least square method. Essentially this algorithm is a fitting of the spectrum. We have added a statement to this sentence: "the retrieval problem can be solved by the linear least-squares such as the optimal estimation method (OEM)".

- **For the "2.1 Retrieval method", suggestions are that 1. Introduce the variables in OEM in combination with the parameters to be inverted; 2. The parameters in the formula are not introduced one by one, such as b, A, I, and Gy.**

Thanks for your suggestion. All the variables and parameters are introduced one by one: "y is the measurement radiance, F is noiseless radiance calculated from the forward model including instrumental response, x is the target atmospheric state vector, $x_a$ is the a priori state vector, b is the parameters in the model that are independent of the state vector" and "where A is the averaging kernel matrix which represent the sensitivity of the retrieved state to the true state and Gy named the contribution function matrix which expresses the sensitivity of the retrieved state to the measurement".

- **In "Since the brightness temperature is proportional to the molecule line intensity and volume mixing ratio (VMR)", the "proportional" is true?**

We are sorry for the mistake here. Because of the saturation of optically thick lines, the statement "the brightness temperature is proportional to … (VMR)" is not always true. The statement has been rephrased in Line 105: "The following molecules are expected to provide useful wind signals based on their spectroscopic line-strengths and typical Earth's VMR: $H_2O$, $O_3$, CO, $O_2$, HF, HCl and O atom (OI)".

- **In "the Doppler shift calculation is performed in the "on-the-fly" mode", can you explain the characteristics of this model?**

The "on-the-fly" mode in ARTS means that each absorption coefficient is obtained from line-by-line calculation of spectral data, rather than extracted from a pre-calculated cross-section look-up table. This means that for each layer of the atmosphere, all the observed frequencies are updated due to the wind and the absorption coefficients are recalculated. This description has been added in Line 114: "Since all the frequencies are updated for each layer of the atmosphere due to the wind and the absorption coefficients also need to be recalculated. To prevent interpolation errors, the radiative transfer calculation is performed in the ARTS "on-the-fly" mode which means that each absorption coefficient is obtained from instant line-by-line calculation".

- **For "The results in Fig. 3 demonstrated that BT induced by wind can be as large as 10 K and different molecules are sensitive to different altitudes", "Although the BT variation (i.e. wind signal) becomes larger with increasing frequency, the system noise temperature also increases correspondingly. From the perspective of SNR, most of the high frequencies do not have better SNR than the low frequencies. The BT of wind at 50km comes mainly from the numerous O3 spectral lines, and the BT induced by wind at 70km is significant at the O2 and H2O spectral lines. Furthermore, wind can be obtained up to 90km by strong spectral lines of O2 and H2O. Finally, BT of wind at 110km exists only at two spectral lines of OI. Therefore, based on the above simulation results", this**

**paragraph needs to be re-expressed, because it is difficult to clearly conclude Table 1 from the above simulation results (Fig. 3), at least to explain the specific basis for the selection of these bands**

Thanks for your comment. The explaination has been rephrased: "It can be seen that no signature of winds are noticeable at 30 km because of pressure broadening of the spectral lines. The number of ΔBT lines at 50 km is the largest and the most dense and comes mainly from the numerous O3 spectral lines. The ΔBT at 70 km is significant at the O2 and H2O spectral lines due to their large VMR or strong line intensity at this altitude region. Furthermore, wind can be obtained up to 90 km by the strong spectral lines of O2 and H2O. Finally, ΔBT at 110 km exists only at 2.06 and 4.74 THz which is the two spectral lines of OI since only this atom has large VMR at this altitude region in THz band molecules. Therefore, based on the above results, those prominent ΔBT positions compared to their surroundings are first selected, such as 118, 448, 487 and 556 GHz lines and so on. Second, other spectral lines that are commonly used or have been mentioned before, even though the ΔBT is not very large, are also taken into account, such as 183, 625 and 773 GHz lines. Third, it is also important to note that since this selection strategy is based mainly on the intensity of the ΔBTs, the O3 line groups with moderate intensity will be missed. Therefore, we have referred to the conclusions of the previous research (Baron et al., 2013a, 2015) such as 359, 655 and 837 GHz line groups and search for the groups of O3 lines with similar density and intensity".

- **"The spectral bandwidth and resolution of the radiometer in this simulation is 4GHz and 1MHz. The system noise temperature for a double sideband (DSB) radiometer at ambient temperature can be simply calculated as a function of frequency (Hubers, 2008)", can you briefly explain the connection between SSB and DSB?**

The THz radiometer uses superheterodyne receiver to obtain spectra. See the figure below, if the target spectrum is the upper band, the SSB is to extract the upper band and eliminate the lower band, while the DSB is to obtain the spectrum after the mixing of the upper and lower band. However, the noise of DSB will be almost the half of SSB.

[Figure]

Figure. Schematic of sideband folding

Reference: Eriksson, P., Ekström, M., Melsheimer, C., & Buehler, S. A. (2006). Efficient forward modelling by matrix representation of sensor responses. International Journal of Remote Sensing, 27(9), 1793-1808.

• **In "Due to the purpose of channel selection, the a priori molecule profiles" which will affect retrieval precision is not considered in this study.", dose the "priori molecule profiles" refer to the accuracy of prior profiles? But the accuracy of the prior profile is included in Sn, how to explain?**

The a priori molecule profiles means the VMR of such as $O_3$, $H_2O$, OI, it is assumed accurate in the simulation and not included in Sn. The a prior error in $S_n$ is from the wind a priori profile, as stated in Sect. 3.2: "The a priori wind profile and corresponding covariance used for retrieval regularization are assumed to be 0 m/s and 100 m/s, respectively".

• **In "Two cases of the same noise (i.e. noise from 0.5MHz resolution) and normal noise" are compared.", how to understand the normal noise?**

As the Eq(10) shown, higher spectral resolution leads to the higher noise. The same noise case means that all the resolution retrievals use the same noise (i.e. the noise is calculated in 0.5 MHz resolution), while the normal noise case means the noise is calculated from Eq(10).

• **Figures 16, 17, 18, etc. do not mark a,b, but a,b is mentioned in the article.**

Thanks for your correction, the marks has been removed.

• **In "The measurement noise induced by receivers is the main error source in OEM retrieval. System noise temperature and NEDT can be calculated according to the (9) and**

**(10)", since the accuracy of the wavelength has a great influence on the inversion accuracy, does the measurement noise include the uncertainty of the wavelength? The accuracy of possible wavelengths also needs to be considered.**

Thanks for your comment. The noise here only consider the thermal system noise (random noise), all systematic errors are not considered. The uncertainty of the frequency is an important systematic error. However, according to the current THz radiometer technology, the variability of the local oscillator frequency is less than 10 kHz which is quite small.

- **For the whole article, does the time resolution impact inversion accuracy? Maybe you can briefly explain the principle of the radiometer in the THz band, and then analyze which factors influence the inversion results.**

The time resolution will not impact the retrieval since a scan time (one profile) of limb sounder is usually less than 1 min. The microwave or THz radiometer system mainly consists of an antenna and scanning drive mechanism, an RF front-end receiver, and back-end digital spectrometers. The atmospheric radiation signal from the antenna is reflected to the receiver system through the main reflector. The first path is the radiation coming in from the antenna, the second is the radiation from the cold space background, and the third is the radiation from the calibration target blackbody. The radiation from the latter two paths is used for the two point calibration. The radiation received by antenna system enters the front-end of the receiver. The signal enters the receiver through feeds of different frequencies, and the front-end signal is transformed into an IF signal through the mixer, and the IF signal is further down-converted after amplification. Then IF down-converter module uses different fundamental frequencies to transform the frequencies to the frequency band suitable for the back-end spectrometers, and these transformed IF signals are fed to the back-end spectrometers and output the power spectrum which can be calibrated to radiance.

The NEDT from system thermal noise is a main factor that affects the retrieval, since it is the random noise which can not be calibrated. As discussed in paper, system noise temperature, spectral resolution and integration time together determine the NEDT. For other parameters, systematic errors from calibration hot-load temperature, radiance linearity assumption, sideband ratio, local oscillator frequency, antenna efficiency, spectroscopic parameters and

LOS azimuth and elevation angles will also impact the retrieval results. However, these factors are instrument-related and require specific analysis which is not the target of this study.

Reference: Baron, P., Murtagh, D., Eriksson, P., Mendrok, J., Ochiai, S., Perot, K., Sagawa, H., and Suzuki, M.: Simulation study for the Stratospheric Inferred Winds (SIW) sub-millimeter limb sounder, Atmos. Meas. Tech., 11, 4545–4566, 2018.

A new section of discussion (Sect. 5) has been added to analyze the general results, limitations and vertical resolution.